# Rapid prediction of key residues for foldability by machine learning model enables the design of highly functional libraries with hyperstable constrained peptide scaffolds

Fei Cai[1], Yuehua Wei[1], Daniel Kirchhofer[1], Andrew Chang[2]*, Yingnan Zhang[1]*

**1** Departments of Biological Chemistry, Genentech, Inc., South San Francisco, California, United States of America, **2** DeepSeq.AI, Inc., San Francisco, California United States of America

* andrew.chang@deepseq.ai (AC); zhang.yingnan@gene.com (YZ)

**Data Availability Statement:** All relevant data are within the manuscript and its supporting information files.

## Abstract

Peptides are an emerging modality for developing therapeutics that can either agonize or antagonize cellular pathways associated with disease, yet peptides often suffer from poor chemical and physical stability, which limits their potential. However, naturally occurring disulfide-constrained peptides (DCPs) and *de novo* designed Hyperstable Constrained Peptides (HCPs) exhibiting highly stable and drug-like scaffolds, making them attractive therapeutic modalities. Previously, we established a robust platform for discovering peptide therapeutics by utilizing multiple DCPs as scaffolds. However, we realized that those libraries could be further improved by considering the foldability of peptide scaffolds for library design. We hypothesized that specific sequence patterns within the peptide scaffolds played a crucial role in spontaneous folding into a stable topology, and thus, these sequences should not be subject to randomization in the original library design. Therefore, we developed a method for designing highly diverse DCP libraries while preserving the inherent foldability of each scaffold. To achieve this, we first generated a large-scale dataset from yeast surface display (YSD) combined with shotgun alanine scan experiments to train a machine-learning (ML) model based on techniques used for natural language understanding. Then we validated the ML model with experiments, showing that it is able to not only predict the foldability of peptides with high accuracy across a broad range of sequences but also pinpoint residues critical for foldability. Using the insights gained from the alanine scanning experiment as well as prediction model, we designed a new peptide library based on a *de novo*-designed HCP, which was optimized for enhanced folding efficiency. Subsequent panning trials using this library yielded promising hits having good folding properties. In summary, this work advances peptide or small protein domain library design practices. These findings could pave the way for the efficient development of peptide-based therapeutics in the future.

**Funding:** The author(s) received no specific funding for this work.

**Competing interests:** The authors have declared that no competing interests exist.

## Author summary

Peptides show promise as therapeutic agents for influencing cellular pathways, but they often lack stability. Disulfide-constrained peptides (DCPs) and de novo designed Hyper-stable Constrained Peptides (HCPs) offer more stable and drug-like modality. Initially, we developed a platform for creating peptide therapeutics using DCPs. However, we recognized the need to improve peptide library design by preserving their ability to fold into stable molecules. We hypothesized that specific patterns in the peptide sequences were vital for proper folding and shouldn't be altered during randomization. To generate effective libraries, we created a method that keeps each scaffold's foldability intact. By combining yeast surface display (YSD) and alanine scanning, we trained a machine-learning model to predict peptide foldability and identify key residues. This model allowed us to design new peptide libraries with foldability optimized. Subsequent tests using this library produced promising results, demonstrating the potential of this method to generate powerful libraries for peptide therapeutic discovery.

## Introduction

Numerous natural peptides have pivotal roles in regulating diverse biological functions, acting as hormones, ion channels, GPCR modulators, growth factors, and neurotransmitters. Typically, they possess selectivity and a strong affinity for binding to cell surface receptors, triggering intracellular signaling cascades. Compared to large biologics, the intermediate size of the peptides, and synthetic tractability and low cost consideration makes them highly attractive as therapeutics modalities [1]. Nevertheless, peptides frequently suffer from poor chemical and physical stability, as well as a short plasma circulating half-life, curtailing their therapeutic efficacy [2]. There are however exceptions, including the naturally occurring disulfide-constrained peptides (DCPs) exhibiting exceptional stability offering alternative strategies to generate drug-like peptides for therapeutic applications [3]. The diverse scaffolds and surface topologies of DCPs render them highly suitable for the generation of large combinatorial DCP libraries to discover new modulators of therapeutically relevant protein targets [4,5,6].

In our previous work, we established a robust platform for discovering peptide leads by integrating DCP scaffold-based libraries with phage display technology. The design of these DCP libraries was guided by insights gained from prior reports and available DCP structures, with a focus on loop flexibility [7]. However, this method does not ensure a dependable design that preserves the native scaffold folding and stability during the randomization of sequences and loop lengths. Consequently, the actual functional members within a given library likely constitute a diminished fraction of the total library population. This limitation contributes to the suboptimal performance of DCP-based libraries. Nevertheless, this platform was successful in generating potent and selective agonists or antagonists, such as a robust agonist for the Wnt pathway [8], a potent antagonist targeting VEGF-A [9], and a highly selective and potent inhibitor of the protease HtrA1 [10]. However, we have observed that in some instances the inherent foldability of natural DCP scaffolds had been compromised due to loop mutations aimed at enhancing affinity, but which turned out to be crucial for retaining the native scaffold and imparting stability to the framework. This trade-off has resulted in tight-binding molecules with poor developability, primarily due to issues encountered in peptide production, such as aggregation and insolubility during the *in vitro* folding step. The process of restoring intrinsic foldability for such peptides is extremely challenging and labor-intensive, impeding drug development.

In addition, we observed a few cases [9] that the top leads from DCP platform adopted a disulfide connectivity different from the cysteine knot of the parent scaffold while maintaining high stability. Therefore, we believe that our existing DCP libraries contain substantial numbers of effective members that adopt completely different conformations from their parent scaffolds.

Therefore, a new design strategy is needed to generate large and diverse DCP libraries, which maintain the intrinsic spontaneous folding property of the parent scaffold. Our aim was to develop a method that allows to predict the pivotal residues that contribute to the spontaneous folding within a given scaffold. These critical residues could then be excluded from randomization in the library design, which should considerably increase the proportion of correctly folded members within the library. To improve diversity, we expanded the library designs to include members that have predicted foldability but may not necessarily maintaining the original knot-like disulfide connectivity and conformations. In order to achieve this goal, we chose a strategy that focused on the amino acid sequences, rather than on 3D structure predictions. Libraries designed by structure-based predictions are restricted to the original cystine-knot scaffold and the currently available programs are unable to accurately predict scaffold conformational rearrangements introduced by changes in disulfide connectivity. Furthermore, our findings indicated a limited relationship between structural modeling outcomes and empirical results (S1 Fig). The discrepancy likely stems from the intricate solvent conditions intrinsic to protein stability and foldability in physiological environments–conditions that present substantial challenges for physics-based simulations.

Previously, Greene LH et al. used site-directed mutagenesis to identify tryptophan as a critical residue for protein folding [11]. We postulated that through a more comprehensive alanine scanning [12], it would be possible to identify more critical residues for a larger range of sequences. However, this kind of experimental approach was very time-consuming and is not a scalable solution for analyzing the foldability profile of a large number of DCP scaffolds.

In this study, we introduce an innovative deep-learning approach to identify pivotal residues in scaffold sequences that are essential for foldability. This method utilizes language model-based techniques, which have great potential to deduce the sequence information harbored within protein and peptide sequences [13,14,15]. By training with data from alanine scanning experiments, we can model peptide foldability with high accuracy. Our methodology diverges from traditional machine learning practices in not just predicting the foldability but in its ability to evaluate individual residues for their importance to foldability. This insight allows us to determine the critical residues for foldability, enabling the swift design of functional peptide libraries.

## Results

### Shotgun Alanine scan as an effective method to generate SAR information between sequences and foldability

Previously, a method combining combinatorial alanine scanning with phage display was developed to swiftly map protein functional epitopes [12] and is a potent tool for generating extensive sequencing datasets, from which structure-activity relationship (SAR) information can be extracted. Here, we applied shotgun alanine scanning to two distinct peptide expression systems, Yeast Surface Display (YSD) [16] and *E.coli*, to generate extensive datasets that could elucidate the correlation between sequence variation and peptide folding propensity. In the YSD system, the exogenous genes were expressed as secreted proteins utilizing the eukaryotic protein production and secretion machinery. Previously, Meng X et al. [17] had developed a method to identify disulfide-rich peptides with robust foldability by leveraging the cellular

protein quality control system in mammalian cells. Likewise, we postulated that any library members capable of successful surface display on yeast cells must be folded into a well-defined 3D structure. Conversely, any unfolded or partially folded members would be subject to degradation through the yeast's rigorous quality control system, aided by its secretion machinery [18]. To confirm this postulation, we generated several control peptide constructs and measured surface display levels in the YSD system. As a positive control, we used the known well-folded peptide 5JI4 containing three disulfide bonds [19]. The negative control peptides, designed to be partially or fully unfolded, were 5ji4CtoA, in which all six cysteines in 5JI4 were mutated to alanine residues; 5JI4N, in which residues of 5JI4 at positions 1, 3, 15–18, 20–22, 32–34 and 36 were mutated to Ala; and SFS, an unrelated linear peptide containing several peptide tags in tandem resulting in a similar length as 5JI4. All peptide constructs had a Myc tag to allow quantification of their yeast surface display levels by fluorescence activated cell sorting (FACS) using a fluorophore-conjugated anti-Myc antibody. As shown in S2A and S2C Fig, the display level of the well-folded positive peptide 5JI4, reflected by the mean fluorescence, was significantly higher than the levels of the three negative controls peptides. This result supported our hypothesis and the signal window between positive and negative control peptides enabled FACS sorting with proper gating for enriching folded vs. unfolded peptides in library screens.

In the *E. coli* system, the peptides were expressed intracellularly and folding was assessed with a modified version of the bimolecular fluorescence complementation BiFC assay [20,21], in which the two non-fluorescent fragments of a fluorescent protein were fused to the peptides' N- and C-termini. If the peptide is folded, then the two fragments will be brought in proximity and become fluorescent. When we fused the two non-fluorescent fragments (YN5ji4TC) to peptide 5JI4, we observed the strong fluorescent signal ('positive'), indicating proper folding. In contrast, when we replaced the coding region of 5ji4 peptide was replaced with a stop codon, a separate RBS (Ribosome Binding Site) and a start codon for the second half of the fragment (YN/TC), we observed fluorescence higher than vector alone but significantly lower than 'positive', which is thus considered as 'background'. In addition, we used the same three negative control peptides described for YSD (5ji4CtoA, 5ji4N and SFS) and made the corresponding pBAD plasmid constructs for the BiFC assay. As shown in S2B and S2C Fig, the 'positive' control peptide gave a strong fluorescence signal, congruent with its expected high foldability. The much lower fluorescence signals of the 'negative' control peptides (S2B and S2C Fig) provided a sufficiently large window to discriminate between the folded and unfolded population with FACS gating.

For testing the shotgun alanine-scanning strategy, we chose HCP (Hyperstable Constrained Peptide) designed *de novo* for exceptional stability [19]. These HCPs were computationally designed with well-defined secondary structures and with variety in topological arrangements, making them particularly susceptible to sequence mutagenesis. We chose HCP with the PDB code 5JI4, designated as 5JI4, to construct alanine-scanning libraries, in which all cysteines were excluded from randomization.

The maximum achievable library diversity using YSD technology and BiFC is approximately $10^8$. By employing degenerate codons, we can confine the mutated amino acids to either two or four at each position [22]. To ensure that the theoretical library diversity aligns with achievable diversity, the total number of amino acids subjected to randomization for each library must not exceed 10–12, which corresponds to a theoretical diversity of ~$10^7$. Consequently, we divided the 5JI4 library into four regions and created four combinatorial alanine scan libraries, as illustrated in Fig 1A and 1B.

For YSD, we constructed the alanine scan library followed by a Myc tag at C-termini as described in a standard protocol [23]. We used anti-Myc antibodies to detect the library

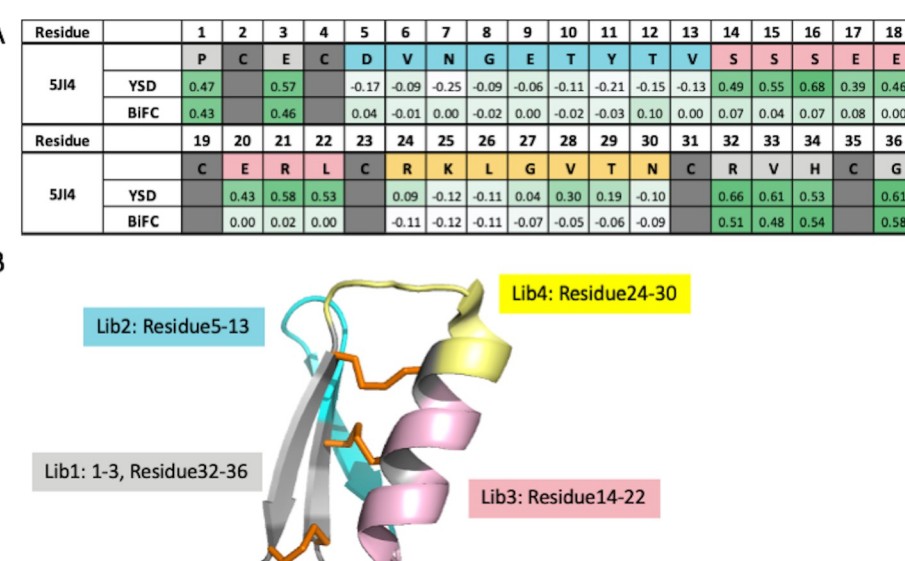

| Residue | | 1 | 2 | 3 | 4 | 5 | 6 | 7 | 8 | 9 | 10 | 11 | 12 | 13 | 14 | 15 | 16 | 17 | 18 |
|---|---|---|---|---|---|---|---|---|---|---|---|---|---|---|---|---|---|---|---|
| | | P | C | E | C | D | V | N | G | E | T | Y | T | V | S | S | S | E | E |
| 5JI4 | YSD | 0.47 | | 0.57 | | -0.17 | -0.09 | -0.25 | -0.09 | -0.06 | -0.11 | -0.21 | -0.15 | -0.13 | 0.49 | 0.55 | 0.68 | 0.39 | 0.46 |
| | BiFC | 0.43 | | 0.46 | | 0.04 | -0.01 | 0.00 | -0.02 | 0.00 | -0.02 | -0.03 | 0.10 | 0.00 | 0.07 | 0.04 | 0.07 | 0.08 | 0.00 |
| Residue | | 19 | 20 | 21 | 22 | 23 | 24 | 25 | 26 | 27 | 28 | 29 | 30 | 31 | 32 | 33 | 34 | 35 | 36 |
| | | C | E | R | L | C | R | K | L | G | V | T | N | C | R | V | H | C | G |
| 5JI4 | YSD | | 0.43 | 0.58 | 0.53 | | 0.09 | -0.12 | -0.11 | 0.04 | 0.30 | 0.19 | -0.10 | | 0.66 | 0.61 | 0.53 | | 0.61 |
| | BiFC | | 0.00 | 0.02 | 0.00 | | -0.11 | -0.12 | -0.11 | -0.07 | -0.05 | -0.06 | -0.09 | | 0.51 | 0.48 | 0.54 | | 0.58 |

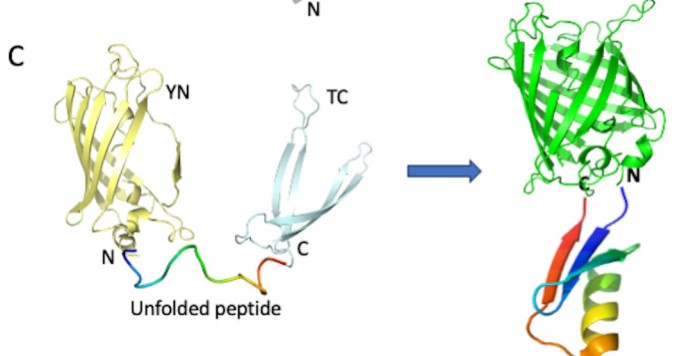

**Fig 1. Shotgun Alanine scanning as an effective method to generate SAR information between sequences and foldability. A.** Sequence representation of alanine scanning libraries (lib1: grey; lib2: blue; lib3: pink; lib4: yellow; no change: dark grey) and alanine scanning results shown as heatmaps of ES for both YSD and BiFC. **B.** Structure of HCP (PDB code 5JI4) with regions that were selected for library constructions indicated in different colors. **C.** BiFC reagent. Half of SYFP2 (light yellow) and half of Turquoise2 (light blue) are fused to the N- and C-terminus of an unfolded peptide (rainbow colors). When peptide is folded, two non-fluorescent fragments will assemble into a green fluorescent complex (green).

population that can be successfully displayed on yeast surfaces and sorted using fluorescence activated cell sorting (FACS). We then perform the next generation sequencing (NGS) on the original alanine scan library and yeast population after one round of YSD sorting. The heat map for enrichment score (ES) was generated after comparing the frequency of a specific amino acid at each position before and after the sorting [7]. In general ES is between -1 to 1, where negative value indicating the de-enrichment and positive value the enrichment. Critical residues important for folding would be enriched after sorting reflected by high positive ES and are colored dark green in the ES heat map (Fig 1A). Residues that are permissive to mutagenesis for folding would have ES close to 0. Residues that were detrimental for folding would be de-enriched after sorting reflected by negative ES. Since the HCP scaffolds were designed for robust foldability with few residues that are detrimental to folding, the ES we observed were mostly in positive range, as expected, with the medium value close to 0.4. Therefore, we consider ES $>= 0.4$ as cut off for residues indispensable for folding.

For the BiFC assay, we constructed the same alanine scan library with an N-terminal fusion of a non-fluorescent fragment SYFP2$_{1-155}$ [24] and a C-terminal fusion of another non-fluorescent fragment Turquoise2$_{156-239}$ [25] (Fig 1C). The expression of the resulting fusion complex is under the tight control of an Arabinose-induced promoter in the special *E. coli* strain SHuffle. A green fluorescent signal was monitored to indicate appropriate peptide folding, and the positive *E. coli* population was sorted using FACS. Similarly, NGS was performed on the cell population before and after sorting, and ES was calculated for each peptide position (Fig 1A).

The results from the YSD method indicated that positions 1–4, 14–23 and 31–36 appeared to be important for folding with average ES score greater than 0.4, while positions of 5–13 and 24–30 were amenable for randomization with ES values close to zero. The results from BiFC showed that the amenable regions are positions 5–30. Both methods identified the N-term (position 1–4) and C-term (position 31–36) regions as critical for folding, which formed the anti-parallel β-sheets in the scaffold. The critical residues identified for folding through shotgun alanine scanning are partially consistent between the two methods with Spearman Correlation 0.77. The major inconsistency is at the positions 14–22, which appear to be amenable in BiFC, but not amenable in YSD. These amino acids are composed of a major part of the α-helix in the parent scaffold. The difference in amenability for this region indicated that yeast has higher stringency for protein folding as previously reported [18], in which peptides with unfolded α-helix cannot be expressed in yeast, whereas partially folded or unfolded α-helix is tolerated in *E. coli*.

The partial consistency for residues identified as critical for folding, and the observed difference in protein quality control regarding α-helix folding between YSD and BiFC suggest that shotgun alanine scanning with yeast surface display is an effective method for establishing the relationship between sequence and foldability for a specific protein domain.

## Data generation for machine learning model with HCP scaffolds

To amass a substantial dataset encompassing information on the relationship between sequence and foldability, we extended the shotgun alanine scan strategy with YSD across seven additional HCP scaffolds having 2–3 disulfide bonds and EETI-II scaffold, reported as the most effective peptide scaffold for library construction [7]. HCP peptides were purposefully designed to represent a diverse range of protein topologies characterized by well-defined secondary structures [19]. The specifics of the libraries for each scaffold's design are depicted in S3 Fig. The library diversities, both theoretical and actual, and the total sequence diversity obtained from NGS are summarized in S1 Table. We screened each library displayed on the yeast surface to select good displayers over one round, and subsequently obtained NGS data for the yeast population before and after sorting.

The screening outcomes are summarized in the ES heatmap presented as "measured" values in S3 Fig and the corresponding bar chart showing the normalized scores are reported in Fig 2. Notably, for scaffolds 2ND2, 2ND3, 5JI4, 5JHI, and gE6 discernible patterns emerge with residues displaying significantly high enrichment scores, as well as regions with relatively low ES. This indicates that these scaffolds are responsive to mutagenesis with regards to folding, and facilitates the identification of essential residues crucial for peptide folding with ES score close to or greater than 0.4. Conversely, scaffolds EETI-II, gEEEH, 5W9F, and 5JG9 only have a limited number of residues that exhibit ES > 0.4, suggesting that these scaffolds are more permissive to mutagenesis in the context of folding.

## Predicting peptide foldability by use of machine learning model

The ultimate goal of this study was to develop a model capable of predicting ES values of individual residues for any given peptide scaffold. However, establishing the ground-truth ES

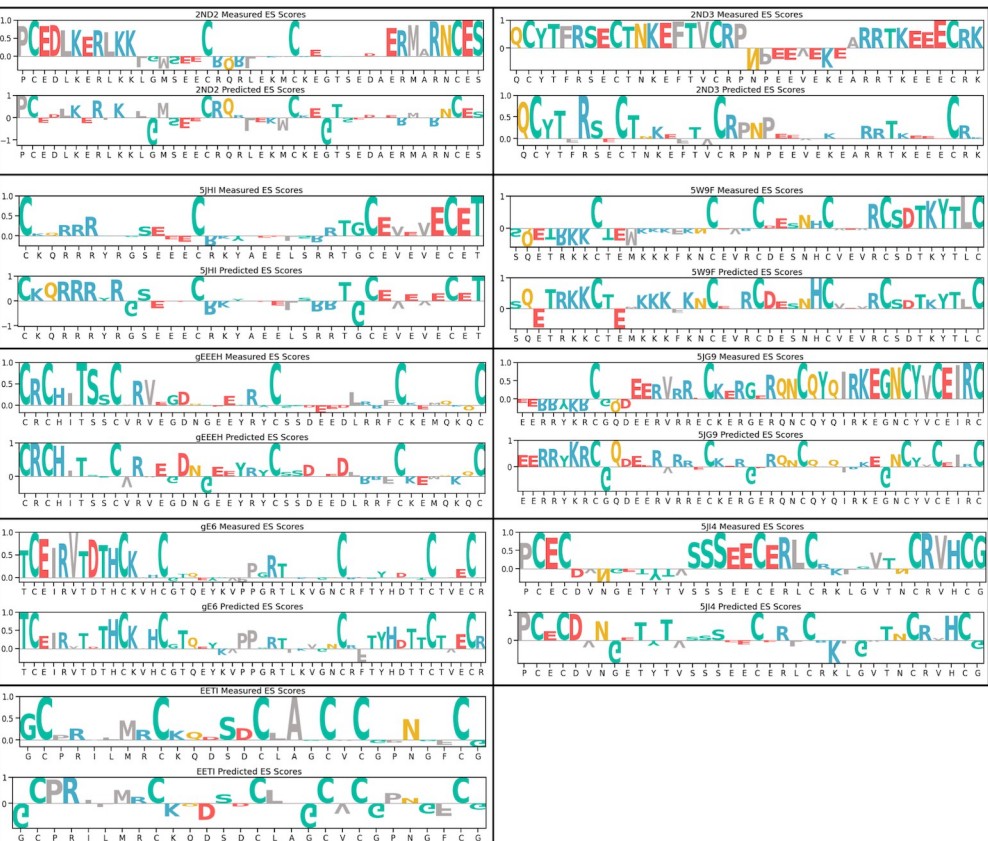

**Fig 2. Sequences and enrichment scores of nine disulfide-rich peptides.** Generation of large datasets from YSD combined with alanine scanning libraries for eight HCP scaffolds and EETI-II scaffold. For each scaffold, both the measured and predicted ES scores were normalized such that the highest score within each scaffold is 1.

values for each scaffold involved summarizing the entire NGS dataset for that particular scaffold, which yielded only one data point per scaffold, insufficient for effective machine learning model training. Therefore, we had to take a detour to reach the goal, in which we first trained the model with classified sequences based on their foldability potential, then we used this trained model to predict ES values indirectly as shown in the next section.

In this study, we hypothesized that specific patterns within peptide sequences contribute to efficient folding and therefore result in high display levels on the yeast surface. The sequences that can be successfully displayed on the yeast surface are defined as 'Displayed Population'. Using the NGS data from the shotgun alanine scan described above, we created a comprehensive dataset of sequences from which we can survey each sequence for their propensity to maintain protein folding. By comparing the abundance level of each sequence from the sorted pools to pre-sort library, we were able to assign labels (high-foldability, mid-foldability, low-foldability) to each sequence.

We performed the following steps for training the model. We first determined the ratio of normalized counts for each sequence in the 'Displayed Population' relative to the original library for each scaffold sample. Secondly, these ratios were categorized into three distinct classes in preparation for training a multi-class classifier. For all scaffolds, sequences exhibiting a ratio exceeding the 95[th] percentile were classified as 'High-foldability', while those falling below the 5[th] percentile were designated as 'Low-foldability'. Sequences with ratios within

these two thresholds were classified as 'Mid-foldability'. For each scaffold, we randomly sampled from the 'Mid-foldability' category to equate its size to the minimum size of either the 'High-foldability' or 'Low-foldability' groups. This approach ensures that each class is well represented during the training phase, facilitating the model's ability to predict foldability across a balanced spectrum of display levels. The specific percentiles were chosen based on our initial search for ideal cutoffs, where we tested several percentile pairs and evaluated classification performance using cross-validation results. This labeled dataset is then used for fine-tuning the language model.

Routinely, F1 score, the fundamental metric for evaluating classification models [26], is used to evaluate the model performance (S2 Table). To refine our evaluation and address the ambiguity associated with the 'Mid-foldability' class, we used 'adjusted F1 score', which simplifies the classification by reducing the three original classes (Low-, Mid-, and High-foldability) to two (Low and Mid-High). Specifically, when the true label indicates low-foldability, predictions of either low or mid are recategorized to 0 (low). However, if the prediction is high, it is reassigned as 1 (Mid-High), indicating a significant discrepancy in prediction accuracy. Conversely, for high-foldability true labels, predictions of mid or high retain the label 1 (Mid-High) to reflect accurate or acceptable predictive performance. If the prediction incorrectly identifies it as low, it is reassigned to 0 (low), highlighting a critical predictive error. For sequences labeled as Mid-foldability, we assign both the true and predicted labels as 1 (Mid-High). This adjustment acknowledges the inherent uncertainty of Mid-foldability sequences, which may exhibit characteristics of both Low and High foldability. This categorization ensures that our model's effectiveness is not overly penalized for these inherently ambiguous cases (S4 Fig). By applying these adjusted labels, we calculate the F1 score, offering a balanced measure of the model's ability in accurately predicting whether a sequence's foldability is low or mid-high.

During each training procedure, we utilized the obtained NGS data from eight HCP scaffolds for training (the "training set") and reserved one scaffold's data for validation (the "validation scaffold"). For example, the adjusted F1 score of 0.966 for 5JHI (Table 1) means using 5JHI as "validation scaffold" while using the other eight scaffolds listed in the Table 1 as the "training set", and we reiterated this operation on all nine scaffolds. This approach enabled us to assess the model's performance in generalizing predictions for novel sequences with low similarity. As depicted in Fig 3, the sequence similarity among the nine scaffolds, determined using the BLOSUM62 matrix, varied significantly, ranging from 16% to 60%.

Our initial NGS data using Miseq Nano kit from Illumina yielded approximately 1,000 unique sequences per scaffold on average, which can be translated to labeled datasets following

**Table 1. Adjusted F1 score of the cross-validation results.**

| Scaffold Name | Adjusted F1 Score (Random) | Adjusted F1 Score (Initial NGS) | Adjusted F1 Score (5X NGS) | Adjusted F1 Score (3 More Scaffolds) |
|---|---|---|---|---|
| 2ND2 | 0.770 | 0.961 | 0.979 | 0.978 |
| 5JI4 | 0.678 | 0.866 | 0.857 | 0.969 |
| 5JHI | 0.780 | 0.966 | 0.987 | 0.968 |
| 2ND3 | 0.792 | 0.790 | 0.979 | 0.924 |
| gE6 | 0.787 | 0.813 | 0.904 | 0.863 |
| EETI-II | 0.764 | 0.701 | 0.732 | 0.841 |
| gEEEH | 0.776 | 0.832 | 0.862 | 0.833 |
| 5W9F | 0.791 | 0.775 | 0.952 | 0.793 |
| 5JG9 | 0.781 | 0.679 | 0.800 | 0.762 |

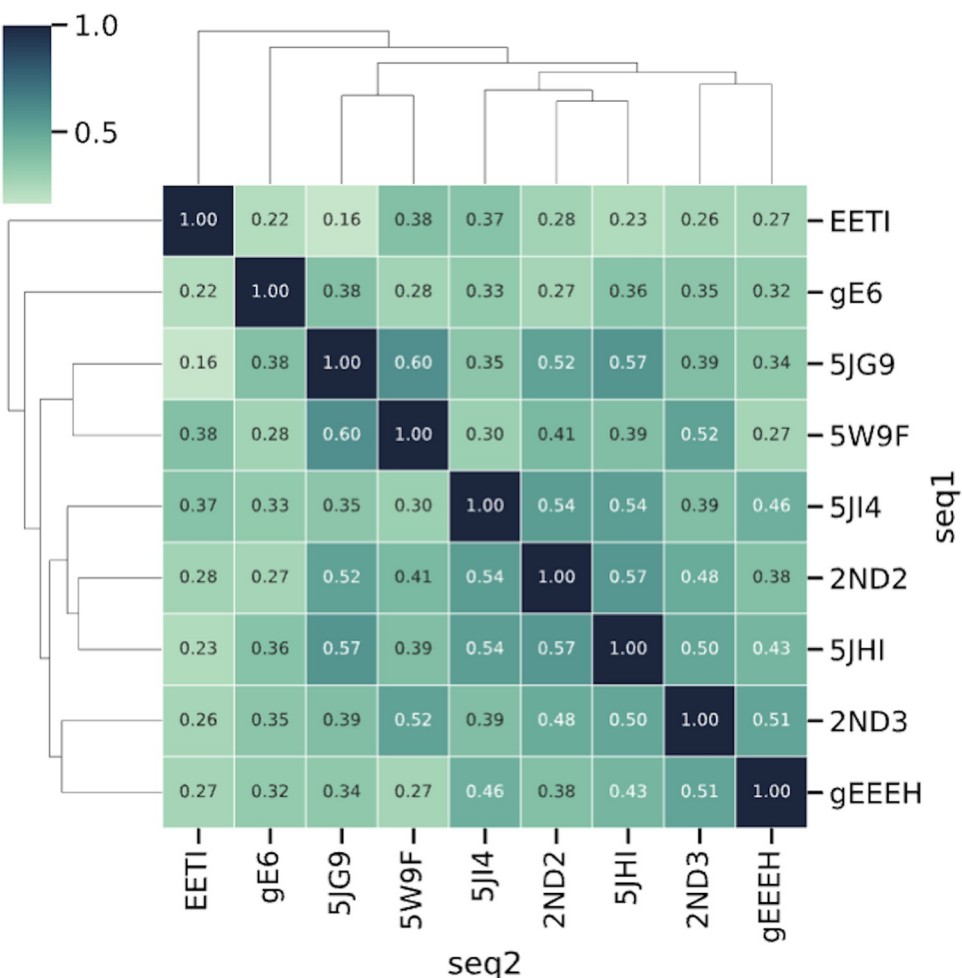

**Fig 3. Sequence similarity among nine scaffolds in the training data.** The figure shows the pairwise sequence similarity among nine HCP scaffolds, as calculated by the BLOSUM62 matrix. The similarity ranges from 16% to 60%, reflecting highly diverse sequence distribution within the training data and their divergence from the validation scaffold sequences. Such a cross-validation approach provides a robust assessment of the model's accuracy and effectiveness in training a robust model that can generalize to sequences away from the training data distribution.

the procedure described above. We evaluated the model's performance using the adjusted F1 score, a metric that measures the model's accuracy while considering the inherent uncertainty in yeast display. The adjusted F1 score, calculated for all nine scaffolds, ranged from 0.68 to 0.97 (Table 1). To determine whether the model performed better than random prediction, we benchmarked it against a model that always predicts the "High-foldability" class. This approach ensures that our model's improved performance is not due to biased guessing and accounts for imbalanced labels. As shown in Table 1, our model fine-tuned on the NGS data outperformed this benchmark, indicating that our model can effectively predict foldability for sequences that exhibit significant differences from the training set, specifically those with less than 60% sequence similarity based on the BLOSUM62 matrix. To further enhance the model's performance, we increased the sequencing depth to cover more reads by using Miseq regular kit from Illumina to generate NGS data, averaging 5,000 reads per scaffold. This resulted in improved adjusted F1 score values ranging from 0.73–0.98 (Table 1).

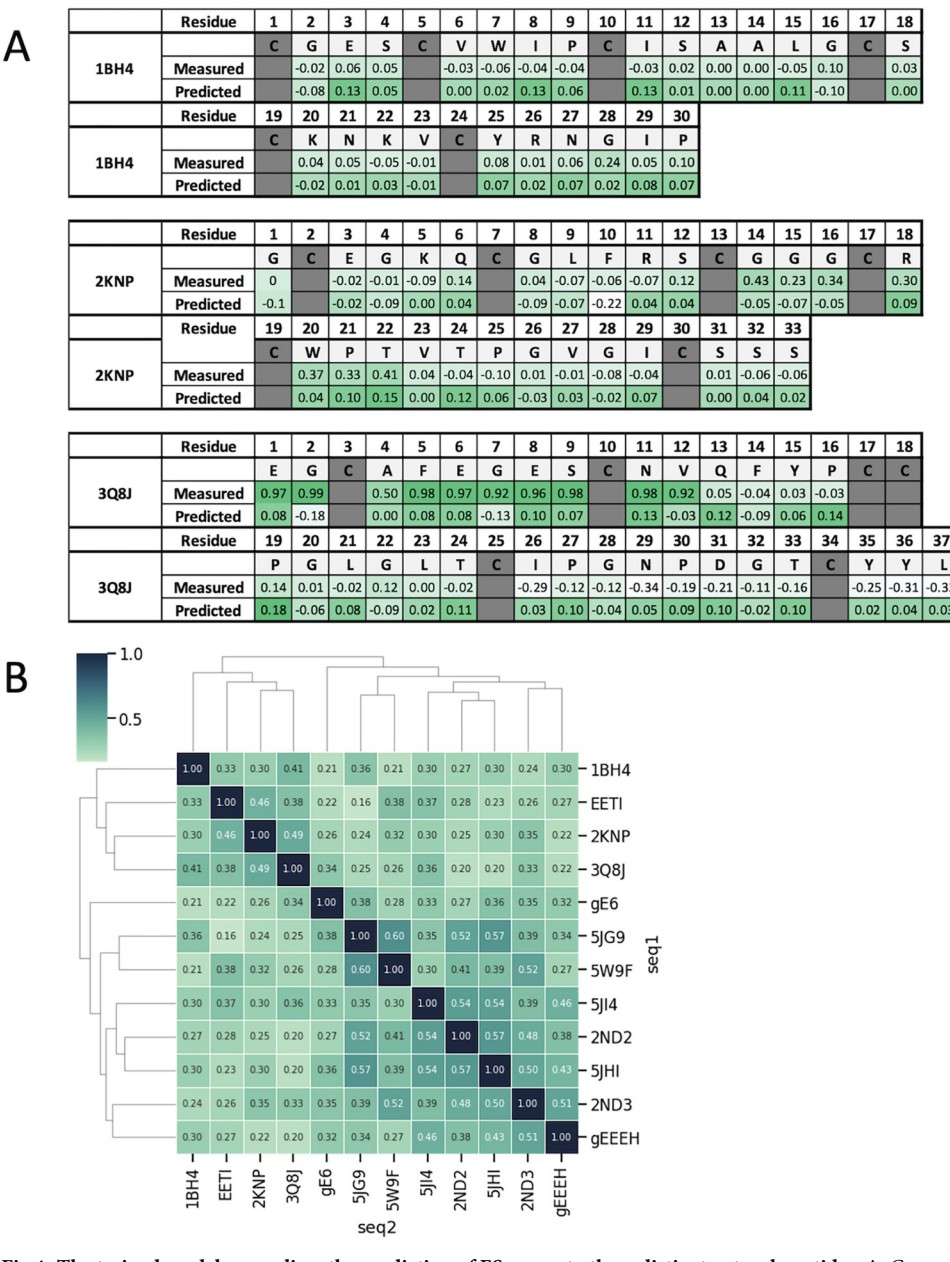

| Residue | 1 | 2 | 3 | 4 | 5 | 6 | 7 | 8 | 9 | 10 | 11 | 12 | 13 | 14 | 15 | 16 | 17 | 18 |
|---|---|---|---|---|---|---|---|---|---|---|---|---|---|---|---|---|---|---|
| | C | G | E | S | C | V | W | I | P | C | I | S | A | A | L | G | C | S |
| **1BH4** Measured | | -0.02 | 0.06 | 0.05 | | -0.03 | -0.06 | -0.04 | -0.04 | | -0.03 | 0.02 | 0.00 | 0.00 | -0.05 | 0.10 | | 0.03 |
| Predicted | | -0.08 | 0.13 | 0.05 | | 0.00 | 0.02 | 0.13 | 0.06 | | 0.13 | 0.01 | 0.00 | 0.00 | 0.11 | -0.10 | | 0.00 |
| Residue | 19 | 20 | 21 | 22 | 23 | 24 | 25 | 26 | 27 | 28 | 29 | 30 | | | | | | |
| | C | K | N | K | V | C | Y | R | N | G | I | P | | | | | | |
| **1BH4** Measured | | 0.04 | 0.05 | -0.05 | -0.01 | | 0.08 | 0.01 | 0.06 | 0.24 | 0.05 | 0.10 | | | | | | |
| Predicted | | -0.02 | 0.01 | 0.03 | -0.01 | | 0.07 | 0.02 | 0.07 | 0.02 | 0.08 | 0.07 | | | | | | |

| Residue | 1 | 2 | 3 | 4 | 5 | 6 | 7 | 8 | 9 | 10 | 11 | 12 | 13 | 14 | 15 | 16 | 17 | 18 |
|---|---|---|---|---|---|---|---|---|---|---|---|---|---|---|---|---|---|---|
| | G | C | E | G | K | Q | C | G | L | F | R | S | C | G | G | G | C | R |
| **2KNP** Measured | 0 | | -0.02 | -0.01 | -0.09 | 0.14 | | 0.04 | -0.07 | -0.06 | -0.07 | 0.12 | | 0.43 | 0.23 | 0.34 | | 0.30 |
| Predicted | -0.1 | | -0.02 | -0.09 | 0.00 | 0.04 | | -0.09 | -0.07 | -0.22 | 0.04 | 0.04 | | -0.05 | -0.07 | -0.05 | | 0.09 |
| Residue | 19 | 20 | 21 | 22 | 23 | 24 | 25 | 26 | 27 | 28 | 29 | 30 | 31 | 32 | 33 | | | |
| | C | W | P | T | V | T | P | G | V | G | I | C | S | S | S | | | |
| **2KNP** Measured | | 0.37 | 0.33 | 0.41 | 0.04 | -0.04 | -0.10 | 0.01 | -0.01 | -0.08 | -0.04 | | 0.01 | -0.06 | -0.06 | | | |
| Predicted | | 0.04 | 0.10 | 0.15 | 0.00 | 0.12 | 0.06 | -0.03 | 0.03 | -0.02 | 0.07 | | 0.00 | 0.04 | 0.02 | | | |

| Residue | 1 | 2 | 3 | 4 | 5 | 6 | 7 | 8 | 9 | 10 | 11 | 12 | 13 | 14 | 15 | 16 | 17 | 18 |
|---|---|---|---|---|---|---|---|---|---|---|---|---|---|---|---|---|---|---|
| | E | G | C | A | F | E | G | E | S | C | N | V | Q | F | Y | P | C | C |
| **3Q8J** Measured | 0.97 | 0.99 | | 0.50 | 0.98 | 0.97 | 0.92 | 0.96 | 0.98 | | 0.98 | 0.92 | 0.05 | -0.04 | 0.03 | -0.03 | | |
| Predicted | 0.08 | -0.18 | | 0.00 | 0.08 | 0.08 | -0.13 | 0.10 | 0.07 | | 0.13 | -0.03 | 0.12 | -0.09 | 0.06 | 0.14 | | |
| Residue | 19 | 20 | 21 | 22 | 23 | 24 | 25 | 26 | 27 | 28 | 29 | 30 | 31 | 32 | 33 | 34 | 35 | 36 | 37 |
| | P | G | L | G | L | T | C | I | P | G | N | P | D | G | T | C | Y | Y | L |
| **3Q8J** Measured | 0.14 | 0.01 | -0.02 | 0.12 | 0.00 | -0.02 | | -0.29 | -0.12 | -0.12 | -0.34 | -0.19 | -0.21 | -0.11 | -0.16 | | -0.25 | -0.31 | -0.33 |
| Predicted | 0.18 | -0.06 | 0.08 | -0.09 | 0.02 | 0.11 | | 0.03 | 0.10 | -0.04 | 0.05 | 0.09 | 0.10 | -0.02 | 0.10 | | 0.02 | 0.04 | 0.03 |

**Fig 4. The trained model generalizes the prediction of ES scores to three distinct natural peptides. A.** Comparison of measured and predicted ES scores for three natural DCPs not included in the model training and independent sample preparation, demonstrating the model's predictive capability. **B.** Sequence similarity between scaffolds used in training and the three new natural peptides (1BH4, 2KNP and 3Q8J), showing notable diversity from the training sequences (with less than 30% average sequence similarity), indicating strong generalizability of the trained model for predicting ES scores from unrelated sequences.

Later, in order to validate the prediction power of current model, we generated three more datasets for three new scaffolds (PDB codes: 1BH4, 2KNP and 3Q8J), that are remotely related to the original nine scaffolds we used for training (Fig 4B). In addition to validation purposes, which will be discussed in the "***Validate the generalizability of machine learning model for foldability prediction***" section of the Results, we incorporated these additional datasets to

further train the model. The results suggested that more diverse datasets could also increase overall model performance, with adjusted F1 values increasing to a range of 0.79–0.98 (Table 1).

However, the foldability prediction for the EETI-II, 5W9F and 5JG9 scaffolds library had relatively low accuracy with initial shallow NGS datasets. We observed that these scaffolds are permissive scaffolds and were not responsive to the perturbance of sequences for folding (Fig 2), suggesting that little sequence signature governing the folding can be deduced. Therefore, it is not surprising that the prediction scores were compromised with these permissive scaffolds (Table 1, initial NGS). Deeper sequencing improved the prediction accuracy for 5W9F and 5JG9 but did not help with the EETI-II scaffold (Table 1, 5X NGS). When we included three more scaffolds with sequence similarity closer to EETI-II, the adjusted F1 score for EETI-II increased while 5W9F and 5JG9 dropped (Table 1).

These results indicate that while the model shows promise for scaffolds that are responsive to sequence perturbation during folding, there is still potential to improve for permissive scaffolds through deeper sequencing and the inclusion of datasets from more diverse scaffolds.

## Trained models can identify pivotal residues for folding

The above models suggest that a given sequence can be used as input to predict whether that sequence will be spontaneously folded, but it cannot predict the influence of individual residues on foldability. Next, we attempted to make the model capable of identifying the important residues responsible for folding.

To achieve this, we employed a method known as Integrated Gradients (IG), which is a method for attributing the prediction of a neural network to its input features (the residues within a given sequence input), quantifying the contribution of each residue to the output decision for a specific class. For a 3-class classification model targeting classes Low-, Mid-, and High-foldability, IG can dissect the influence of each residue of within a given sequence on the probability assigned to the 'High-foldability' class, elucidating how changes in residues impact the classification outcome. In natural language processing, IG has been applied to tasks such as sentiment analysis, text classification, and machine translation [27,28,29]. In genetics, similar methods have been utilized to identify DNA sequence motifs, contributing to our understanding of gene regulation and function [30,31,32]. More recently, Wang Z et al. have employed IG to identify key residues in protein-target binding, further demonstrating the versatility of this approach [33].

Using the IG method, we systematically calculated the predicted ES value based on the attribution score for each residue for all nine scaffolds. The results are summarized in Fig 2 as 'Predicted' and are compared to the actual ES scores measured by alanine scanning experiments ('Measured') side by side. Since the model was not trained to learn the actual ES scores, the predicted ES scores may be on a different scale than the actual ES scores.

To assess the accuracy of the ES values predicted by the Integrated Gradients (IG) method, we evaluated the model's ability to predict non-touchable residues. Non-touchable residues are defined as those with ES values equal to or greater than the average ES value within the given scaffold, which are then set to one, while other residues are set to zero (S5 Fig). We calculated the accuracy score for each scaffold by determining the percentage of correctly classified residues (both touchable and non-touchable) identified by the IG method from the trained machine learning model, compared to those determined through statistical calculations from the NGS data (Table 2). Additionally, we benchmarked this performance against a random model that always predicts every residue as non-touchable. This approach ensures that our model's improved performance is not due to biased guessing and effectively accounts for imbalanced non-touchable and touchable residues.

**Table 2. Accuracy of non-touchable residue prediction measured ES scores vs. predicted ES scores.**

| Scaffold Name | Accuracy (Random) | Accuracy (Initial NGS) | Accuracy (5X NGS) | Accuracy (3 More Scaffolds) |
|---|---|---|---|---|
| 5JHI | 0.400 | 0.714 | 0.800 | 0.743 |
| gEEEH | 0.366 | 0.780 | 0.756 | 0.805 |
| gE6 | 0.383 | 0.553 | 0.745 | 0.766 |
| EETI | 0.393 | 0.750 | 0.750 | 0.679 |
| 5JI4 | 0.556 | 0.611 | 0.639 | 0.694 |
| 2ND2 | 0.488 | 0.674 | 0.698 | 0.674 |
| 5W9F | 0.415 | 0.659 | 0.561 | 0.537 |
| 2ND3 | 0.763 | 0.447 | 0.526 | 0.526 |
| 5JG9 | 0.600 | 0.511 | 0.467 | 0.533 |

In this study, the initial machine learning model was trained to classify peptide sequences based on their display levels on yeast surface: high-, mid-, or low-foldability. The model has not learned to assign ES values to individual residues within each sequence. Thus, calculating predicted ES values for each residue represents a complex extension of the model's initial capabilities. Although the IG method provides a solution to this complexity that enhances the model's analytical scope from mere classification of overall foldability to a detailed residue-specific prediction of expression potential, it could introduce inaccuracy in prediction due to such extension.

Our analysis showed strong ES score prediction for most scaffolds except 2ND3 and 5JG9, with accuracy scores below random model prediction. This inaccuracy persisted even when the model was enriched with data from three additional scaffolds. Notably, 5JG9 has the lowest accuracy scores in direct foldability classification (Table 1), suggesting that indirectly predicting ES scores adds a layer of complexity to the model, which could result in a certain degree of inaccuracy. Further detailed investigation is required to fully understand and improve the model's performance in these instances.

Despite these complexities, the ability of our model to achieve such accuracy was promising, suggesting that we can use this model to identify crucial residues that contribute to folding in any given peptide sequences, as long as they are within the length range as our training datasets, which are between 20–50 amino acids long with either two or three disulfide bonds.

## Validate the generalizability of machine learning model for foldability prediction

Firstly, in an effort to assess our model's capability to generalize the prediction of critical residues contributing to foldability on entirely unseen and non-similar sequences, we selected three DCP scaffolds derived from natural peptides with PDB code of 1BH4, 2KNP and 3Q8J. These were markedly distinct from the HCP sequences utilized in our training data (sequence similarity is less than 30% on average compared to the training data) (Fig 4B). We challenged our model to compute the ES values for each residue within each scaffold and then compared these predictions with the real experimental data obtained from shotgun alanine scans with YSD (Fig 4A). The accuracy for predicting non-touchable residues reached an average of 0.67 when the model was trained on deeper sequencing data, which is above the average accuracy of a random model prediction of 0.4 (Table 3). This result affirms our model's ability to predict ES values across diverse protein sequences. This result showed that our trained model can successfully generalize to new distant natural peptides.

**Table 3. Accuracy of non-touchable residue on three new scaffolds from natural peptides.**

| Scaffold Name | Accuracy (Random) | Accuracy (Initial NGS) | Accuracy (5X NGS) |
|---|---|---|---|
| 1BH4 | 0.333 | 0.767 | 0.700 |
| 2KNP | 0.424 | 0.667 | 0.697 |
| 3Q8J | 0.432 | 0.432 | 0.622 |

The second method to validate the model prediction power is to directly evaluate the peptide display level on yeast surface for the libraries designed by our machine learning model. We chose two scaffolds with PDB code of 5JI4 and 3Q8J and generated "positive" yeast-display libraries, in which amenable positions were randomized with degenerate codon of NNK and critical positions for folding (non-touchable) predicted by our ML model unchanged. We also created "negative" libraries for each scaffold by randomizing the predicted non-touchable positions, excluding cysteines, while keeping amenable residues unchanged. The peptide display levels on yeast surface of these libraries were quantified by FACS as the mean fluorescence of the histogram curves (Fig 5). If the predicted non-touchable residues are accurate, we should see higher display level of the "positive" libraries than their counterpart "negative" libraries. The results showed that the "positive" libraries yielded higher peptide display levels, outperforming the "negative" libraries by 1.8 and 3.2 folds for 5JI4 and 3Q8J, respectively, confirming the accuracy of the prediction.

Additionally, we used the current ML model to evaluate a previously developed DCP platform for the discovery of high affinity peptide ligands [7]. At that time, library design was largely based on existing reports and insights derived from available structural information.

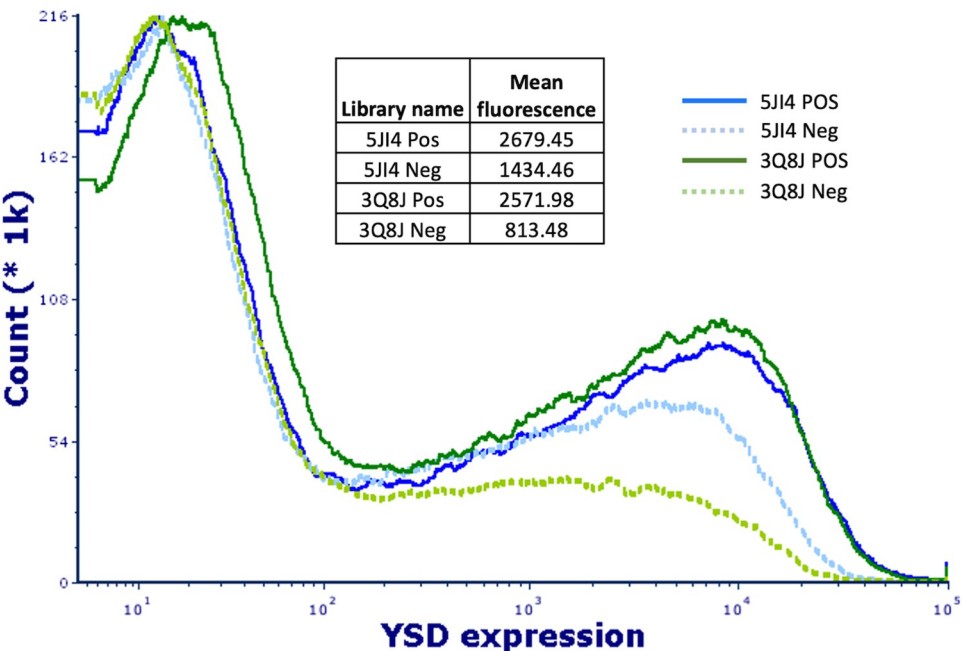

**Fig 5. Validation of machine-learning-predicted non-touchable residues for folding using yeast surface display.** The libraries are constructed based on an HCP scaffold and a DCP scaffold, with PDB codes of 5JI4 (blue curve) and 3Q8J (green curve), respectively. Solid lines: positive libraries with the randomization on amenable residues; dashed lines: negative libraries with randomization on "non-touchable" residues.

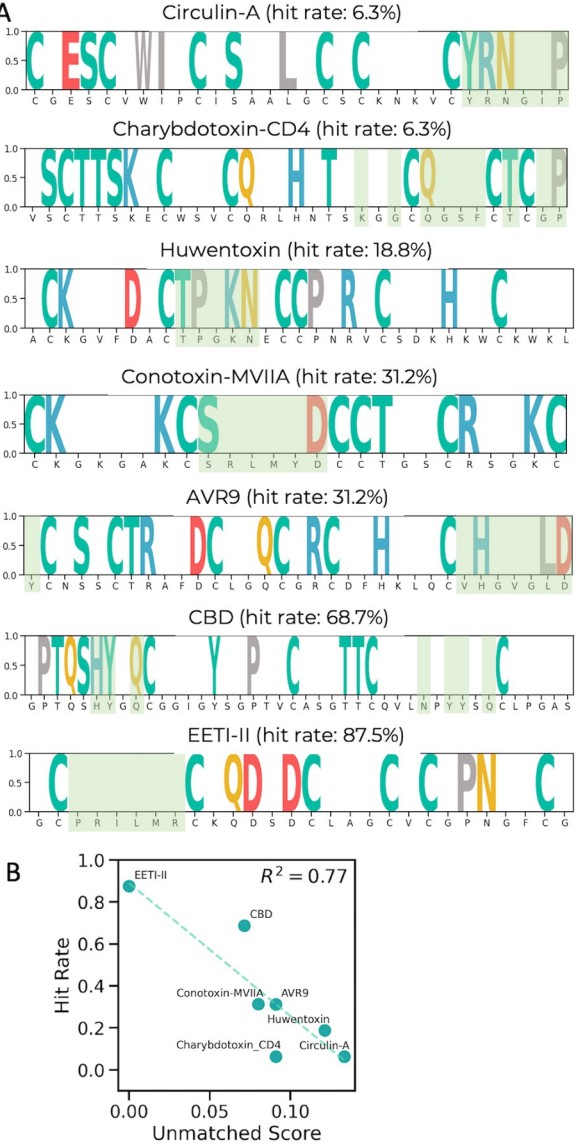

**Fig 6. Re-evaluation of DCP libraries reveals design insights for improved hit rates. A.** ES score distribution across various DCP scaffolds with the originally randomized regions marked in green colored regions. **B.** Correlation between the "unmatched scores" and the hit rates for 7 previously reported DCP scaffolds; a higher degree of mismatch corresponds to lower hit rates, emphasizing the importance of model-guided design.

When we assessed the effectiveness performance of these libraries, we observed a notable variability among different scaffolds [7]. This performance was measured by the hit-rate, the ratio between the number of successful panning campaigns from which validated binders had been identified to the total number of panning campaigns conducted.

With the introduction of our current AI model, we re-evaluated these DCP libraries and identified certain shortcomings in the initial design. We computed the 'non-touchable' residues by identifying those with ES scores above the average ES score within each scaffold. Residues at these positions had predicted ES scores below the average, with those scores being reset to zero, while the remaining residues, classified as non-touchable, were assigned a value of one (Fig 6A). The randomization regions chosen in prior studies are highlighted in green colored

boxes. As depicted in Fig 6A, most of the randomization residues (green colored regions) chosen for the Circulin A and Huwentoxin libraries in the prior study were predicted as nontouchable by our current model, corresponding to relatively low hit rates (6.3%, and 18.8%, respectively).

Conversely, the library designs for AVR9, Conotoxin-MVIIA, CBD, and EETI-II ([7] and Fig 6A) were in close alignment with the predictions of our AI model. In these cases, the randomization regions largely overlapped with the predicted regions having low foldability scores. As a result, the hit rates for these libraries ranged from moderate to high (31.2–87.5%).

To quantify this relationship, we introduced an 'unmatched score' defined by the formula:

$$unmatched\ score = \frac{number\ of\ mismatched\ residues}{length\ of\ the\ scaffold}$$

Mismatched residues are those within the previously randomized regions that our model predicts as having an importance above the scaffold-wide average. Fig 6B demonstrates a strong correlation between the unmatched score and hit rates, further confirming the predictive accuracy of our AI model.

The above two types of experiments and the re-evaluation of previously reported peptides scaffolds libraries with our AI-model validated our model's predictive power and illustrated its practical utility in designing stable libraries. By identifying the critical residues contributing to foldability, our model can guide the construction of more effective peptide libraries by increasing the percentage of foldable population in library pool.

## Evaluate effectiveness of an HCP library (5JI4) designed with the consideration of foldability

To demonstrate the superiority of the library constructed while considering foldability, we developed a new library using the 5JI4 scaffold and assessed its effectiveness. Using the updated model trained with expanded 12 datasets (nine original scaffolds plus three additional DCP scaffolds), we identified the following positions on 5JI4: 6, 8–9, 11, 13, 17–18, 20, 22, 24–28, 30, 33 and 36, as amenable for randomization (Fig 7A). Residues at these positions had predicted ES scores below the average, with scores below this threshold reset to zero, while the remaining residues were classified as non-touchable and set to one, as shown in Fig 7A. These positions roughly matched the amenable region 1 (position 5–13) and region 2 (position 24–30) identified by alanine scan experiments, in which residues had ES score < 0.4 (S6 Fig), by an accuracy score of 0.7 (Table 2). Informed by these insights, we designed two libraries with hard randomization in the amenable regions 1 and 2 identified from our Alanine scan experiments (Fig 7B, green-colored regions), respectively, which were displayed on major coat protein p8 of M13 phage. The diversity achieved was 2.5 x $10^9$ for 5JI4-Lib1 and 5 x $10^9$ for 5JI4-Lib2.

To determine the performance of the libraries for ligand discovery we chose a panel of eight protein targets, which have no structural homology and a wide mass range (Fig 7C). We screened these eight targets with solution panning strategy [34] and two additional plate panning [35] for CD8α/β and CD28, thus ten panning campaigns in total. We consider clones showing spot phage ELISA signal > 0.3 and signal/noise ratio > 2 as positive hits. The total hit numbers for each panning campaign are summarized in Fig 7C (detail in S3 Table). At least one binding hit was discovered for nine out of the ten panning campaigns, surpassing the success rate of the best DCP scaffolds reported previously [7]. We also obtained the NGS data for these screening and the ranking of clones for their enrichment along the rounds of screening was analyzed and considered the top-ranking clones as potential hits. Based on their sequence

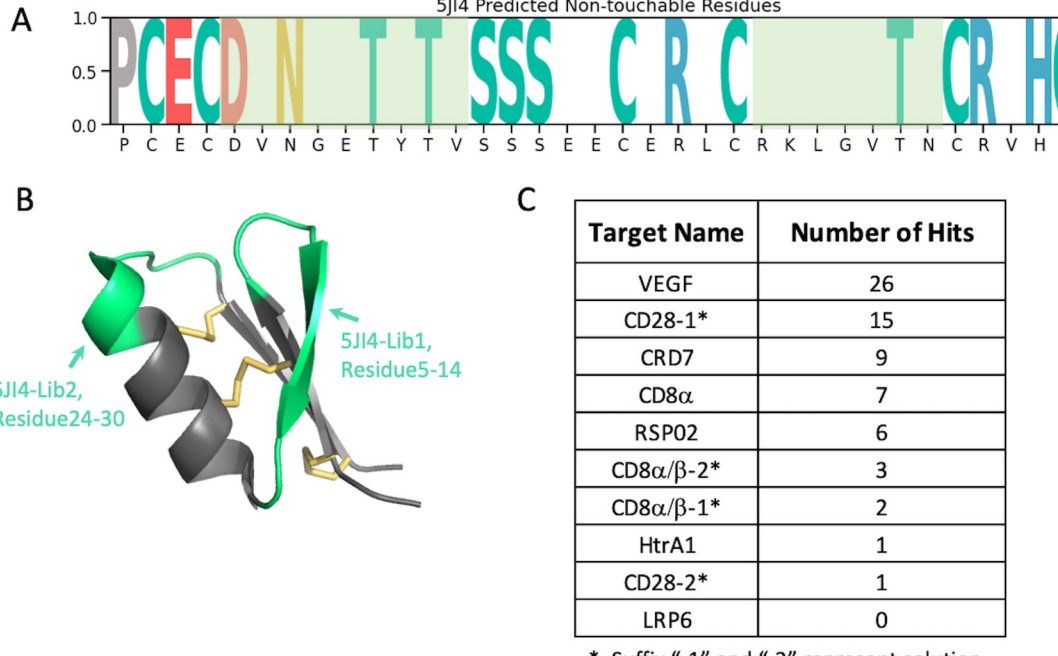

**Fig 7. Validation of newly constructed 5JI4 phage library based on the predicted ES scores. A.** Visualization of 5JI4's ES scores, where the positions scoring below the median are set to zero. The randomization regions are colored with light green. **B.** 5JI4 structure with randomized regions highlighted in green. **C.** Summary of the hit numbers achieved by the 5JI4 phage library for ten panning campaigns.

diversity we selected 16 potential hits spanning five protein targets for peptide synthesis in their linear form. Seven of these are hits with positive phage ELISA signal and nine were selected based on NGS ranking without spot ELISA available. Subsequently, we conducted *in vitro* folding experiments on these peptides in various buffers. We observed a strong propensity for spontaneous folding regardless of the buffer condition. All HCP hits tested successfully folded into three-disulfide-containing product within two days in a universal buffer system (PBS, pH 7.4, 50% DMSO) without the need for a redox pair, as shown in Fig 8A. The binding of these folded peptides to their respective target proteins was confirmed using SPR (Table 4 and S1 Data) and binding kinetics are exemplified by the high-affinity peptide 5ji4-CRD7-2 (see Fig 8B). Among seven selected ELISA-positive hits, all of them were successfully synthesized and *in vitro* folded with three disulfide bonds and five were proven to be validated binders by SPR (Table 4). Nine peptides selected from the NGS ranking were synthesized and folded with the expected number of disulfide bonds and five of them bound to their target with affinities in the μM range as determined by SPR (Table 4). In general, binding affinities for these SPR-validated initial hits are weak, with $K_d$ values in the micromolar range and fast on- and off-rates. There are two exceptions with binding affinities in the nanomolar range. One of these two, 5ji4-CRD7-2, has a measured $K_d$ of 30 nM (Fig 8B), and the other one, 5ji4-VEGF-3, has $K_d$ of 120nM.

The results indicate that a library constructed with foldability considerations is highly effective, demonstrating the necessity of foldability prediction for library design based on a large number of peptide scaffolds. Our machine learning model for foldability prediction with satisfactory accuracy will be of great utility in this context.

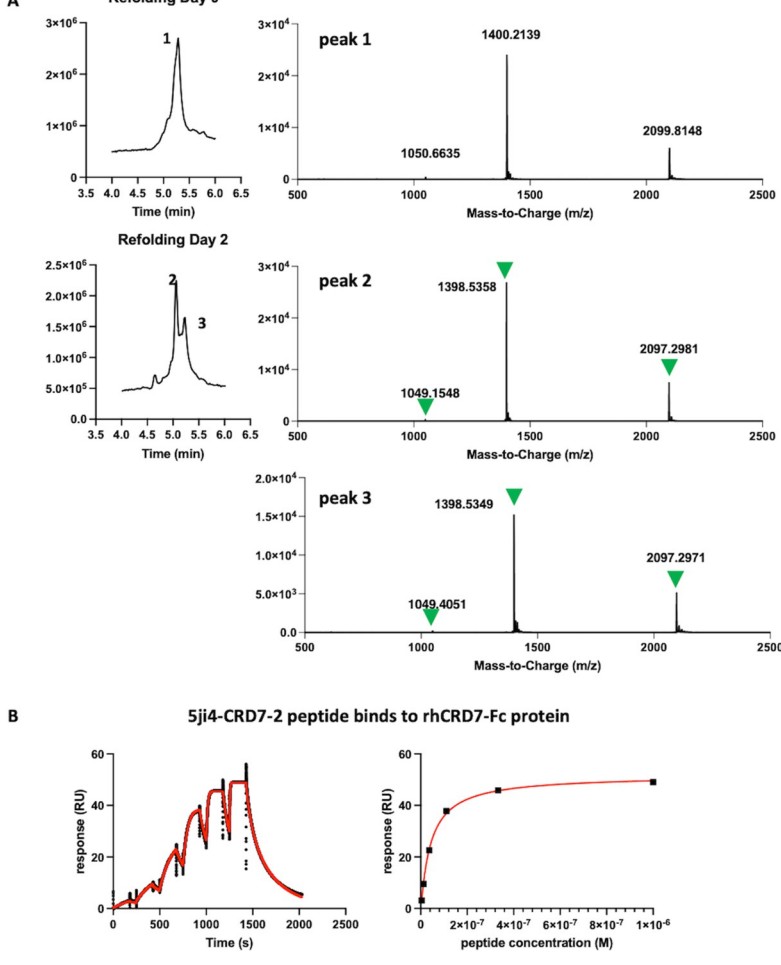

**Fig 8. Confirm the outcome of the hits from 5JI4 libraries. A.** An example of peptide 5ji4-RSP-1 *in vitro* re-folding. TIC from LC/MS runs before and after re-folding are shown on the left with peaks' m/z on the right. After two days of re-folding, both peaks 2 and 3 contain expected m/z (indicated by green triangles) for folded peptides with three disulfide bonds. **B.** An example SPR sensor gram for measuring peptide binding affinity. Kinetic fitting is on the left and steady-state affinity is on the right. Experimental data and fitting curves are shown in black and red, respectively.

## Identification of functional inhibitors of the protease HtrA1

The availability of a sensitive enzymatic assay allowed us to test the potential inhibitory activity of 5ji4-A7, the sole binding hit for HTRA1 (Figs 7C and 9A and S1 Data). HTRA1 is a trypsin fold serine protease implicated in various diseases, including Alzheimer's disease, cancer, arthritis and age-related macular degeneration [36,37,38,39,40,41]. The enzymatically active for of HTRA1 is a homotrimer with three symmetrically arranged active sites [42,43]. This binding hit and the parent scaffold peptide 5ji4-A7 were synthesized and refolded and tested in the HtrA1 enzyme assay. As shown in Fig 9B, 5ji4-A7 inhibit HtrA1 activity in a concentration-dependent manner with an IC$_{50}$ value of 38 μM, whereas the parent 5ji4 was not inhibitory. The complete inhibition by 5ji4-A7 suggested that all three active sites of the HTRA1 trimer were neutralized by the peptide.

An AF2-predicted structure of 5ji4-A7 superimposed with the solution NMR structure of the parent scaffold 5ji4 (PDB 5JI4) is shown in S6 Fig. We noticed that the disulfide connectivity remained unchanged (1 disulfide is not formed in the AF model) and that the structures

**Table 4. Affinity summary for selected positive hits.** 16 clones were selected for synthesis and *in vitro* folding, of which 7 clones are with spot ELISA signal range in 0.25–4.0 and signal/noise (s/n) ratio between 2–40 and nine clones were selected based on NGS ranking. The $K_d$ of synthetic peptides were measured by SPR with steady state or kinetic fitting.

| | | ELISA | | SPR |
|---|---|---|---|---|
| Name | sequence | signal | s/n | $K_d$ (μM) |
| **5ji4** | PCECDVNGETTYTVSSEECERLCRKLGVTNCRVHCG | | | |
| **CD8α** | | | | |
| 5ji4-CD8-2 | PCECVWESFAHLPWSSEECERLCRKLGVTNCVCG | | | 4.2 ± 4.3 |
| 5ji4-CD8-8 | PCECLVTPYKDDYKSSEECERLCRKLGVTNCRVHCG | | | N.B. |
| 5ji4-CD8-16 | PCDCDVNGETYTVSSEECEPLCLNWLSSLCRVHCG | 3.43 | 37.34 | N.B. |
| 5ji4-CD8-20 | PCECDVNGETYTVSSEECERLCVWWLSTVCRVPCG | | | 23 ± 7* |
| 5ji4-CD8-21 | PCECVWESFAHLPWSSEECERLCLNWLSSLCRVHCG | | | 6.0 ± 2.2 |
| 5ji4-CD8-22 | PCECLVTPYKDDYKSSEECERLCVWWLSTVCRVPCG | | | 76.3 ± 35.7* |
| **CD28** | | | | |
| 5ji4-CD28-4 | PCECEDTTFFIIFNSSEECERLCRKLGVTNCRVHCG | | | N.B. |
| 5ji4-CD28-7 | PCECDVNGETYTVSSSEGCERLCSWDQGKWCRVHCG | 0.25 | 3.48 | N.B. |
| 5ji4-CD28-13 | PCECEDTTFFIIFNSSEECERLCSWDQGKWCRVHCG | | | 32.3 ± 12.2* |
| **VEGF** | | | | |
| 5ji4-VEGF-2 | PCECFMYDILVGLASSEECERLCRKLGVTNCRVHCG | 4.71 | 44.54 | 6.8 ± 3.6 |
| 5ji4-VEGF-3 | PCECDVNVETYTVSSSEECKRLCWGFGQSDCRVHCG | 3.07 | 32.18 | 0.12 ± 0.03 |
| 5ji4-VEGF-5 | PCECFMYDILVGLASSEECKRLCWGFGQSDCRVHCG | | | 86 ± 11* |
| **CRD7** | | | | |
| 5ji4-CRD7-2 | PCECPIRPYGVPVKSSEECERLCRKLGVTNCRVHCG | 0.37 | 4.60 | 0.03 ± 0.01 |
| 5ji4-CRD7-3 | PCECDVNGETYTVSSEECGRLCMNDWATTCRVHCG | 0.65 | 9.52 | 122 ± 74* |
| 5ji4-CRD7-5 | PCECPIRPYGVPVKSSEECERLCMNDWATTCRVHCG | | | N.B. |
| **RSP02** | | | | |
| 5ji4-RSP-1 | PCECDVNGETYTVSSEECEPLCWMYNEHFCRVHCG | 0.58 | 5.22 | 22 ± 17 |

*Steady state $K_d$; N.B: no binding

superimposed well with each other with an RMSD value of 0.750 Å. The main difference lies in the region that had been mutated (residue 5–14) that formed antiparallel beta sheets in original scaffold. The residues that had been identified as untouchable (residue 15–27) are mainly involved in α-helix formation and were kept almost unchanged in the predicted structure with an RMSD of 0.257 Å. Therefore, in the case of 5JI4, the sequences that are responsible for α-helix formation are the main determinants for foldability, corresponding to high ES in Ala-scan experiment as well as the high score from our ML model prediction.

We then performed affinity maturation of 5ji4-A7 following the strategies described before [7]. Library screening yielded five hits with improved spot ELISA binding signals (Fig 9A) and they were chemically synthesized followed by *in vitro* folding. Their binding affinities, determined by SPR, were improved up to 7-fold compared to the 5ji4-7A ($K_d$ = 2.72 μM) (Fig 9A). In HtrA1 enzyme assays, the five peptides showed 3-10-fold improved inhibitory potencies and they all inhibited HtrA1 activity by 100%, indicating that all three HtrA1 active sites were neutralized (Fig 9).

Our study demonstrates that the libraries designed based on 5JI4 scaffold possesses the ability to produce functional inhibitors to protease HtrA1, with the primary hit showing potential for enhancement through an effective affinity maturation strategy.

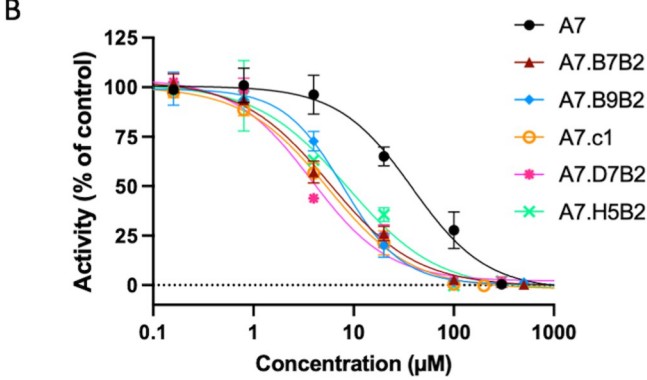

**A.**

| Name | Sequence | ELISA | | SPR | IC50 |
|---|---|---|---|---|---|
| | | signal | s/n | Kd,μM | μM |
| 5ji4 | PCECDVNGETYTVSSSEECERLCRKLGVTNCRVHCG | 0.83 | 1.4 | n/a | n/a |
| 5ji4_A7 | PCECTILRGWHFLTSSAECERLCRKLGVTNCRVHCG | 3.36 | 5.8 | 2.72 | 38.3 ± 7.9 |
| 5ji4_A7.B7B2 | PCECSMLPGWTFMTSSAECERLCDRLGLRTCRVHCG | 3.54 | 7.2 | 1.65 | 5.6 ± 0.8 |
| 5ji4_A7.B9B2 | PCECTILPGWHFLTSSAECARLCDRLGLRTCRVHCG | 3.67 | 7.0 | 0.37 | 7.8 ± 0.6 |
| 5ji4_A7.D7B2 | PCECSILRGWHFLTSSAECERLCDRLGLRTCRIHCG | 3.45 | 5.3 | 0.85 | 3.7 ± 0.2 |
| 5ji4_A7.H5B2 | PCECTILPGWYFLTSSAECERLCDRLGLRTCRVHCG | 3.59 | 7.6 | 0.44 | 11.4 ± 3.1 |
| 5ji4_A7.c1 | PCECSILPGWHFLTSSAECARLCDRLGLRTCRIHCG | 3.55 | 7.5 | 0.48 | 5.7 ± 1.3 |

**Fig 9. Functional and selective inhibitors against HtrA1 generated from 5JI4 libraries. A.** Sequences for HtrA1 primary hit and affinity maturation hits with Spot ELISA, SPR measured $K_d$ and $IC_{50}$ measured by enzymatic assay. **B.** Inhibition of HtrA1 enzyme activity by HCP derived from 5JI4 libraries.

## Discussion

For decades, it has been recognized that the linear sequence of amino acids dictates the protein folding process. However, understanding how the physicochemical properties encoded in this 1D amino acid sequence determine the 3D structure of a protein has posed a challenge that scientists have long sought to unravel [44,45,46]. Early investigations employing a variety of biophysical and computational techniques [47] identified simple sequence patterns that may facilitate folding [11,48]. Subsequently, more complex determinants, such as secondary structures and their overall topological arrangement [49] were discovered to contribute to protein folding dynamics. Yet, translating these discoveries into computational algorithms capable of predicting 3D folding based solely on primary sequences remained elusive until recent years.

The emergence of machine learning models has revolutionized this landscape, enabling the prediction of a protein's 3D structure from its primary sequence with unprecedented accuracy [50,51]. However, while these tools excel at predicting exact 3D structures, they often overlook foldability—the inherent propensity of a linear polypeptide chain to spontaneously fold into a stable 3D conformation. When constructing libraries based on small protein or peptide scaffolds, accurate prediction of exact 3D structures may not be essential. Instead, rapid evaluation of foldability across a large number of sequences is crucial for enriching the library with well-folded members. In this context, a sequence-based machine learning model presents an ideal solution to address this challenge.

We have devised experiments capable of generating extensive datasets that capture the relationship between sequence and foldability. By combining alanine shotgun scanning with yeast surface display, we generated this dataset and utilized it to train a machine learning model. This model was then employed to predict foldability for various peptide scaffolds. To validate

the predictions, we conducted two distinct experiments: first, by extending the prediction to peptide scaffolds that are remotely related to those used in the training dataset, and second, by confirming that peptides predicted to have good foldability by our ML model exhibited higher display levels on the yeast surface compared to those predicted to have poorer foldability. Additionally, we further confirmed the predictive accuracy of our AI model by reassessing the performance effectiveness of a panel of previously reported DCP-based peptide libraries [7] (Fig 9).

We subsequently developed a novel HCP library, 5JI4, based on experimentally validated foldability predictions. The library was meticulously designed to maximize the presence of well-folded members within the naïve pool. Remarkably, the library's efficacy rivaled that of the best scaffold, EETI-II, reported previously when libraries were constructed without foldability considerations. Furthermore, the identified hits exhibited enhanced *in vitro* folding efficiency, and functional inhibitors against protease HTRA1 were successfully obtained. Notably, this library is built upon an entirely *de novo* designed peptide scaffold [19]. Our findings indicate that this comprehensive artificial library is comparable to previously reported disulfide constrained peptide platforms that were based on natural peptide scaffolds [7].

Protein design methods leveraging structure prediction, such as ProteinMPNN [50], are also instrumental to design protein with higher foldability. A fundamental distinction between ProteinMPNN and our approach lies in the underlying assumptions about the resulting 3D structure of the generated sequences. ProteinMPNN operates on the premise that the newly derived sequences will retain the original scaffold's 3D structure. In contrast, our method is not bound by this limitation, thus enabling the generation of a broader array of sequences and permitting a more varied and adaptable 3D conformation. This capacity for greater diversity is particularly advantageous given our objective to create extensive libraries. The versatility in sequence and structure can potentially elevate the probability of identifying foldable and effective hits.

In this context, we deployed our model to produce ten unique sequences, based on the EETI-II scaffold, that pinpoint regions suitable for randomization and residues likely to maintain foldability. We also generated ten sequences using ProteinMPNN for comparative purposes, with a sampling temperature of 0.2 and certain cysteine positions held constant [2,9,15,19,21, and 27]. Upon evaluating sequence variability, our model exhibited a heightened degree of diversity (a lower average sequence similarity score) compared to the sequences derived from ProteinMPNN (Fig 10A and 10C). Furthermore, structural predictions performed via AlphaFold2 revealed that our sequences not only adopt distinct conformations but also exhibit a degree of structural flexibility (Fig 10B and 10D).

It is important to note that our model primarily predicts foldability, which is the propensity of a linear polypeptide to fold into a well-defined 3D structure, rather than predicting the actual structure itself. For a sequence derived from a specific scaffold, there is a high likelihood that it may adopt a completely different structure or conformation compared to its parent scaffold, as long as it has a high propensity to fold. Consequently, the library designed based on such predictions would exhibit significant conformational diversity, with substantial structural variations from the parent scaffolds, including differences in disulfide bond connectivity for disulfide constrained peptides.

Despite the promising prediction power of our machine learning model, it does have some limitations. Due to the complexity extension from the original model that was trained on the classification datasets, the accuracy of pinpointing the "non-touchable" residues for folding is far from perfect. To address this, we employed the Integrated Gradients attribution method to interpret what the trained model has learned about these 'non-touchable' residues. AI explanation is an evolving field, and several other attribution methods could be further explored.

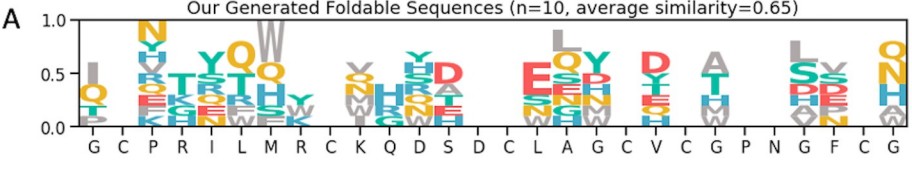

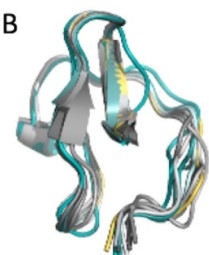

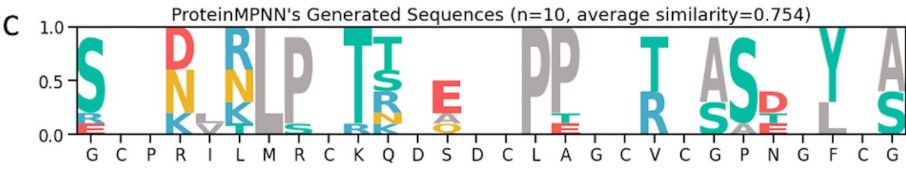

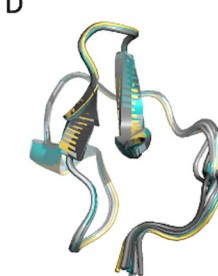

**Fig 10. Comparison of sequence and structure diversity between our method and ProteinMPNN. A.** Sequence logo plot from ten unique sequences generated by our model based on the EETI-II scaffold. **B.** Predicted 3D structures of the ten sequences from our model, displaying some conformation variability. **C.** Sequence logo plot from ten unique sequences generated by ProteinMPNN based on the EETI-II scaffold. **D.** Predicted 3D structures of the ten sequences from ProteinMPNN, showing limited conformation variability.

However, we have seen the trend of improved prediction accuracy when the model was trained on deeper sequencing data as well as trained with more diverse scaffolds that are remotely related to the original nine scaffolds used early in this study (Tables 1 and 2). Therefore, the model holds potential to be further improved with more datasets from a large range of diverse scaffolds. Future sensitivity analyses could determine the optimal sequencing depth and diversity to improve the data collection later.

On the other hand, when it comes to library construction, a degree of imprecision in foldability prediction may be acceptable. The key metric lies in the notable enhancement of well-folded members within the naïve library compared to a random library constructed without foldability considerations. In this context, the breadth of prediction efficacy across a wide range of sequences outweighs the need for pinpoint accuracy. Specifically, we observed that inaccuracies tend to occur more frequently with scaffolds that are permissive to mutagenesis for foldability. Typically, libraries based on such permissive scaffolds exhibit satisfactory effectiveness even when constructed without considering foldability, as demonstrated by the case of EETI-II in a previous report [7]. Hence, the influence of inaccurate predictions on permissive scaffolds is relatively minimal in terms of library effectiveness.

Current model was trained by the data generated on peptide scaffolds with 28–47 residues long and containing 2–3 disulfide bonds. Therefore, we only have the confidence to apply the model to the peptide or small protein domains that fall into the similar molecular weight range (2-5Kd) containing similar number of disulfide bonds. To predict foldability for larger proteins, the model needs to be trained with more datasets based on the proteins of larger size and further validation is needed.

In summary, the AI model developed herein enables rapid identification of crucial residues for folding within a peptide scaffold sequence, facilitating the strategic design of randomization regions to maximize the proportion of well-folded members in the library. Foldability

predictions can be generated within minutes for a large batch of sequences, allowing the swift design and construction of library ensembles encompassing thousands of diverse peptide scaffolds representing a broad spectrum of sequences and structures. Furthermore, compared to protein design methods reliant on structure prediction, our methodology, centered solely on foldability prediction, broadens the scope of achievable structural diversity. This approach paves the way for the construction of peptide libraries based on an unlimited array of scaffolds, whether *de novo* designed or derived from natural sources, exhibiting a rich diversity in structure and conformation. Such libraries hold great potential as robust platforms for peptide therapeutic discovery.

## Methods

### Yeast surface display and FACS sorting

The standard protocol [23] was followed to construct the YSD display library and perform FACS sorting. Briefly, oligomers with degenerated codons for alanine scanning were ordered through either IDT (Integrated DNA Technologies, Inc., Coralville, IW) or Gene Link (Gene Link, Inc., Elmsford, NY) with 50 bp of homology at each end to the pCTCON2 vector, flanking *NheI* and *BamHI* sites, for homologous recombination. Degenerated oligomers were PCR amplified with universal primers YDrecF (5'-AGT GGT GGA GGA GGC TCT GGT GGA GGC GGT AGC GGA GGC GGA GGG TCG GCT AGC-3') and YDrecR (5'-TGT TAT CAG ATC TCT ATT ACA AGT CCT CTT CAG AAA TAA GCT TTT GTT CGG ATC C-3') for 25 cycles using 2X CloneAmp HiFi master mix (TAKARA, #639298), following the manufacturer's protocol. PCR products were column purified and eluted in dI-water according to the manufacturer's manual (QIAGEN #28104). pCTCON2 vector was cut with *NheI-HF* and *BamHI-HF* (NEB #R3131 & #R3136, respectively), treated with CIP (NEB #M0525), then gel purified and eluted in dI-water (QIAGEN #28506). Subsequently, PCR-amplified inserts and double-digested pCTCON2 vector were co-transferred into *S. Cerevisiae* str. EBY 100 following the updated electroporation protocol [16]. FACS sorting was done following the standard protocol [23] but only with anti-Myc labeling (primary chicken IgY anti-c-Myc, Invitrogen, #A-21281; secondary Alexa Fluor 488-goat anti-chicken IgG, Invitrogen, #A-11039) using a BD FACS sorter Aria II Fusion with purity mode. Yeast cells before and after sorting were sampled and kept at -80˚C until ready for NGS.

### BiFC assay and FACS sorting

For the BiFC assay, a positive control was generated first as follows. pBAD/Myc-HisB vector (Invitrogen, #44001) was digested with *NcoI-HF* and *HindIII-HF* (NEB, #R3193 & #R3104, respectively), treated with CIP, then gel purified and eluted in dI-water. A gBlock encoded the designed fusion protein SYFP2$_{1-155}$-GS-5JI4-GS-Turquoise2$_{156-239}$ (Fig 1C) was ordered from IDT, with 15 bp of homology at each end to the pBAD/Myc-HisB vector, flanking *NcoI* and *HindIII* sites. Cloning was done by using an In-Fusion HD cloning kit (TAKARA, #102518) following the manufacturer's manual. The resulting plasmid pBAD-YN5ji4TC was confirmed with SANGER sequencing and used as a template to generate the negative control by replacing the 5ji4 region with follows: a stop codon followed by a *BglII* site, an RBS site, an *EcoRI* site, and a start codon. Quikchange PCR was performed with primers TCquikFWD (5'-TAA GAT CTA GGA GGA ATT CAT GGA CAA ACA GAA AAA TGG TAT AAA GG-3') and YNquik-REV (5'-GAA TTC CTC CTA GAT CTT ATT AGG CAG TGA TGT ACA CAT TAT GG-3') using QuikChange II XL Site-Directed Mutagenesis Kit (Agilent, #200521). The resulting plasmid pBAD-YN/TC was confirmed with SANGER sequencing and used for negative controls.

Alanine scanning library for BiFC assay was prepared similarly as for the YSD except for the listed differences. Oligomers with degenerated codons have 15 bp of homology at each end to the pBAD-YN/TC plasmid, flanking *BglII* and *EcoRI* sites. Degenerated oligomers were PCR amplified with universal primers YN3'fwd (5'-CCA TAA TGT GTA CAT CAC TGC CGG-3') and TC5'rev (5'-CCT TTA TAC CAT TTT TCT GTT TGT CTG AAC C-3'). pBAD-YN/TC plasmid was linearized with *BglII* and *EcoRI-HF* (NEB, #R0144 & #R3101, respectively). An In-Fusion HD cloning kit was used and the reaction mixture was directly transferred into SHuffle Express *E. coli* cells (NEB, #C3028J) for library construction. One positive control and two negative controls were generated by transferring plasmids pBAD-YN5-ji4TC, pBAD-YN/TC, and empty vector pBAD/Myc-HisB to SHuffle cells, respectively.

For FACS sorting, libraries and controls are incubated at 30°C until OD600 reaches 0.4~0.8. Then 0.02% (v./v.) $_L$-arabinose was added to each culture and expression was induced at 25°C for 3 h. After induction, 2 O.D. of cells was taken from each culture for FACS sorting, monitoring green channel. *E. coli* cells before and after sorting were sampled and kept at -80°C until ready for NGS.

## Next Generation Sequencing (NGS)

NGS was performed as described in a previous study [7] with some modification at the first PCR step depending on different starting materials. For phage samples, the eluted phage from panning was used directly as a template in the first NGS PCR. For *E. coli* samples, plasmids were purified from each sample and approximately 20 ng was used in the first NGS PCR. For yeast samples, 1 μl of yeast culture was used with Phire Plant Direct PCR Master Mix (Thermo, #F160S) in the first NGS PCR. MiSeq Reagent Nano Kit v2 and MiSeq Reagent Kit v2 were used in this study.

## NGS data processing

Raw paired-end amplicon sequencing reads were first merged using VSEARCH. The merged reads were then processed through an internal Python pipeline. In brief, it first extracted the unique molecular identifiers (UMIs) and the initial or final nine bases, depending on the primer used. Following UMI extraction, DNA sequences were translated into protein sequences. The pipeline quantified each unique protein sequence by counting the distinct UMIs associated with it, which was then used as a measure of copy number. These counts were subsequently normalized against the total read volume for each sample, enabling standardized comparisons across the samples. The ranking of the individual clones is based on the normalized copy numbers. Sequences not matching the original design specifications within each sample were identified and excluded from subsequent analysis.

## Residue Enrichment Score (ES) calculation

To evaluate the significance of each residue within a scaffold sequence regarding its impact on stability and foldability, we proposed that essential residues for maintaining the sequence's structure will appear more frequently at specific positions in the expressed population compared to their occurrence in the original shotgun alanine scan library. Therefore, we computed a weighted ratio that contrasts the frequency of each residue in the expressed population with that in the original library. This ratio, termed the Enrichment Score (ES), serves as an index of each residue's contribution to the foldability of the scaffold sequence.

Let $F_{exp}$ represent the Expressed Frequency per residue, and $F_{orig}$ represent the Original Frequency per residue. ES for each residue can be defined by the following conditional

equations:

$$ES = \begin{cases} \dfrac{F_{exp} - F_{orig}}{1 - F_{orig}} & if\ F_{exp} > F_{orig} \\[2ex] \dfrac{F_{exp} - F_{orig}}{F_{orig}} & otherwise \end{cases}$$

## Model training

In this study, we employed the RoBERTa model [52] which utilizes a robustly optimized Bidirectional Encoder Representations from Transformers (BERT) pretraining approach and has been pre-trained on the MGnify dataset [53]. To tailor the model to our specific task of predicting peptide expression outcomes, we augmented the pre-trained network with an additional dense layer followed by a projection layer.

During the fine-tuning process, all model layers were kept trainable to ensure comprehensive updating of parameters. We used cross-entropy as the loss function and employed the AdamW optimizer to adjust the weights. The learning rate was set to $1\times10^{-5}$, with a weight decay of 0.01 and a dropout rate of 0.1 to mitigate overfitting. The fine-tuning was conducted over 10 epochs.

## Evaluating model performance on the classification task

To ensure that our model can accurately predict expression and foldability for sequences it has not encountered, we applied a rigorous cross-validation framework. Following the model's training on processed data from eight scaffolds, we performed inferencing on the ninth scaffold's sequences, data that was held back from the model during training. This is to assess the model's generalization capability. To accommodate the intrinsic variability present in cellular expression systems, we employed the adjusted F1 score metric computed with the standard operation [26].

## Critical residue identification and evaluation

To examine the specific amino acids influencing a scaffold's expression, we utilized the integrated gradients technique. This advanced feature attribution method pinpoints the significance of each input feature for predictions made by neural networks. To ensure robust generalizability of attributions, we adopted a stringent cross-validation scheme as described in the classification task. The model, trained on datasets from eight distinct scaffolds, excluding the scaffold currently under prediction, was used to assess each residue's contribution to the expression outcome.

Integrated gradients, as described by Sundararajan et al. [27], determine the gradient of the model's prediction output in relation to each individual input feature, in this case, each amino acid residue. The approach integrates these gradients from a baseline input, representing a neutral or 'null' state, to the actual sequence input. This method systematically uncovers the significance of each residue throughout the sequence, providing a detailed perspective on its influence on expression levels.

Here, we computed attribution scores for residues in relation to the 'High-Expression' category to pinpoint amino acids essential for protein expression. These scores were contrasted with the ground-truth enrichment score. To assess our model's predictive accuracy for these important residues, we tested its ability to predict non-touchable residues. Non-touchable

residues are those with ES values equal to or greater than the average ES value within a specific scaffold (see S5 Fig). The accuracy score was then calculated for each scaffold by comparing the non-touchable residues identified by the IG method in the trained machine learning model with those determined through statistical calculations from the measured NGS data.

## Protein reagents

The following proteins were purchased commercially: CD8α-Fc (R&D, #AVI10927), CD8α/β (R&D, #9358-CD-50), Fz7CRD-Fc (R&D, #6178-FZ-50), CD28-Fc (R&D, #342-CD-200), and RSP02-Fc (Biolegend, #786206). Rest proteins that were used in this study were produced in-house (Genentech). They are listed as following in the format of "Protein name (construct, DNA ID, expression system)": VEGF$_{8-109}$ (A27-D191, DNA1AE76566, *E. coli*), LRP6_E1-E2 (A20-E631, DNA681820, Tni Baculovirus), and HtrA1$^{PD-S/A}$ (D161-K379:S328A, DNA713424, *E. coli*).

## Phage library construction and panning

The 5ji4 libraries were constructed following the Kunkel mutagenesis method [54] and the solution panning protocol was followed as previously described [34]. After four to five rounds of binding selection, while eluted pool was subjected to NGS analysis, individual phage clones were also picked and analyzed with phage spot ELISA. Positive clones were subjected to SANGER sequencing.

For the target HtrA1, after initial panning, five primary hits from 5JI4-Lib1 were identified, one from SANGER sequencing and four from NGS ranking. Among these five hits, A7, which is the SANGER hit, was picked for affinity maturation. Following the same strategy [7], one soft- and one hard-randomization library was designed targeting the 5JI4 variable regions 1 and 2, respectively. Two libraries were each panned as previously described [7]. Both phage spot ELISA coupled with SANGER sequencing and NGS analysis were applied after affinity maturation.

## Peptide synthesis and *in vitro* folding

Peptide synthesis was done by CSBio (Menlo Park, CA). The linear peptides were synthesized as free N- and C- terminus. For the small-scale *in vitro* folding test, linear peptides were dissolved in 100% DMSO as 10 mg/ml stock, then 1/20 diluted into folding buffers. Five different buffer conditions were tested: [1] 0.1 M NH$_4$HCO$_3$ pH 9.0, 2 mM reduced glutathione, 0.5 mM oxidized glutathione, and 9% DMSO; [2] 0.1 M NH$_4$HCO$_3$ pH 8.0, 1 mM reduced glutathione, and 55% DMSO; [3] dI-water w. 5% DMSO; [4] 1X PBS pH 7.4 w. 10% DMSO; and [5] 1X PBS pH 7.4 w. 50% DMSO. Refolding was promoted by incubation at R.T. with gentle rotation and monitored by LC-MS for disulfide bond formation. HCP peptides with three disulfide bonds were observed in all tested buffer conditions, regardless of the presence or absence of a redox pair. Therefore, buffer condition [5] was used as a universal refolding buffer for all HCP peptides. After refolding, peptides with three disulfide bonds were purified by HPLC according to standard procedure and subjected to SPR measurements.

## Surface Plasma Resonance (SPR)

All binding experiments were performed using a Biacore S200 instrument (GE Healthcare) at 25˚C at a flow rate of 30 μl/min in HBS-EP buffer (Cytiva, #BR100188) w. 5% DMSO. The kinetics and affinity measurements were carried out using single-cycle kinetics (SCK) and referenced by subtracting the signal from the blank flow cell. Binding affinity and kinetic

parameters (association and dissociation rate constants, and dissociation equilibrium constant) were analyzed in a three-fold dilution series starting from approximately 100 μM, then calculated with the Biacore S200 Evaluation Software (version 1.1.1, Cytiva) using either a 1:1 binding model or a steady-state equilibrium model (according to the manufacturer's instructions). Sensor Chip Protein A (Cytiva, #29127555) was used for proteins CD8α-Fc, CD28-Fc, Fz7CRD-Fc, and RSP02-Fc, while Series S Sensor Chip SA (Cytiva, #29104992) was used for CD8α/β, VEGF$_{8-109}$ and HtrA1$^{PD-S/A}$.

## HtrA1 enzymatic assay

For enzymatic assays we used human full-length HtrA1 (HtrA1), which was expressed in Trichoplusia ni cells and purified as described [42,55]. Enzyme assays with the fluorescence-quenched synthetic substrate Mca-IRRVSYSF(Dnp)KK (H2-OPT) [42,56] were carried out essentially as described [42]. HCP stock solutions in DMSO were diluted in 50 mM Tris-HCl, pH 8.0, 200 mM NaCl, 0.25% CHAPS (assay buffer) and incubated with HtrA1 in 96-well black optical bottom plates for 15 min at 37˚C. A 10 mM stock solution of H2-OPT in DMSO was diluted with water, prewarmed at 37˚C and then added to the HCP-enzyme mixtures. The final concentrations in the reaction mixtures were: 1 nM HtrA1, 5 μM H2-OPT, 0.1% DMSO and varied concentrations of HCP. Substrate cleavage was measured at 37˚C on a SPECTRA-max M5 microplate reader (Molecular Devices) and the initial rates of cleavage determined. The IC50 values were determined by fitting the data to the 4 parameter sigmoid function.

## Supporting information

**S1 Fig. Structural based stability prediction does not correlate with the actual yeast display result.** To determine the free energy change upon protein unfolding (ΔΔG), we employed software provided by Cyrus Biotechnology, which utilizes the Rosetta suite for structure prediction. We initiated ΔΔG calculations from the relaxed and energy-minimized structure of the 5JI4 scaffold to ensure the reference conformation was at the lowest possible energy state. We mutate each residue position, with the exception of cysteine residues due to their role in disulfide bond formation, to all amino acids except cysteine. The ΔΔG values were computed for each mutated residue per position. Furthermore, the mean ΔΔG across all mutations per position was calculated, providing an aggregate measure of stability changes attributable to each position. In our analyses, a ΔΔG value falling below the median for the entire scaffold was indicated with an 'X' in the corresponding table, denoting a residue crucial to folding stability. Conversely, in the context of Yeast Surface Display (YSD), a residue's Enrichment Score (ES) surpassing the median was marked as 'X', highlighting its significance.
(PPTX)

**S2 Fig. Positive and negative controls for FACS sorting of folded vs. unfolded peptides.**
**(A) YSD.** Yeast cells with no expression (Neg., black line), or expressing peptide 5JI4 ("positive"; green line), 5ji4CtoA (no disulfide bond control; orange line), 5ji4N (Alanine substitution of key residues; red line), or SFS (linker control; blue line) resulted in different distribution pattern in flow cytometry histograms. The grey arrow bar indicated the gating used for FACS sorting. **(B) BiFC.** E. coli cells harboring the empty vector (Neg., black line), pBAD-YN/TC ("background"; gray line), pBAD-5ji4CtoA (no disulfide bond control; orange line), pBAD-5ji4N (Alanine substitution of key residues; red line), pBAD-SFS (linker control; blue line), and pBAD-YN5ji4TC ("positive"; green line) are well-separated in a flow cytometry histograms. **(C)** Sequences of control peptides and mean fluorescence for each construct

measured using YSD and BiFC.
(PPTX)

**S3 Fig. Sequences and enrichment scores of nine disulfide-rich peptides.** Generation of large datasets from YSD combined with alanine scanning libraries for eight HCP scaffolds and EETI-II scaffold. The same color code as in Fig 1A for library designs and ES heat maps. Showing both measured and predicted ES scores.
(PPTX)

**S4 Fig. Illustration of adjusted label and confusion matrix using the entire cross-validation datasets (9 scaffolds combined).**
(PPTX)

**S5 Fig. An example of the definition of non-touchable residues.**
(PPTX)

**S6 Fig. Superimposing the structure of 5JI4 (PDB 5ji4, yellow) and the Alphafold2-predicted structure of HtrA1-A7.**
(PPTX)

**S1 Table. Library diversity and total sequence diversity for all scaffolds.**
(XLSX)

**S2 Table. Raw cross-validation results of each multi-class classifier trained on 5X NGS data.**
(XLSX)

**S3 Table. All sequences and phage spot ELISA data for the table summarized in Fig 6C.**
(XLSX)

**S1 Data. LC-MS data for all synthetic peptides reported in Table 4 and Fig 9A.**
(PPTX)

## Author Contributions

**Conceptualization:** Fei Cai, Andrew Chang, Yingnan Zhang.

**Data curation:** Fei Cai, Yuehua Wei, Daniel Kirchhofer, Andrew Chang, Yingnan Zhang.

**Formal analysis:** Fei Cai, Yuehua Wei, Daniel Kirchhofer, Andrew Chang, Yingnan Zhang.

**Investigation:** Daniel Kirchhofer, Andrew Chang, Yingnan Zhang.

**Methodology:** Fei Cai, Yuehua Wei, Andrew Chang, Yingnan Zhang.

**Project administration:** Fei Cai, Daniel Kirchhofer, Andrew Chang, Yingnan Zhang.

**Resources:** Andrew Chang, Yingnan Zhang.

**Software:** Andrew Chang.

**Supervision:** Daniel Kirchhofer, Andrew Chang, Yingnan Zhang.

**Validation:** Fei Cai, Andrew Chang, Yingnan Zhang.

**Visualization:** Andrew Chang, Yingnan Zhang.

**Writing – original draft:** Fei Cai, Andrew Chang, Yingnan Zhang.

**Writing – review & editing:** Fei Cai, Daniel Kirchhofer, Andrew Chang, Yingnan Zhang.

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
