## [Decision Letter · Decision Letter 0]

12 Aug 2024

Dear Dr. Zhang,

Thank you very much for submitting your manuscript "Rapid prediction of key residues for foldability by machine learning model enables the design of highly functional libraries with hyperstable constrained peptide scaffolds" for consideration at PLOS Computational Biology.

As with all papers reviewed by the journal, your manuscript was reviewed by members of the editorial board and by several independent reviewers. In light of the reviews (below this email), we would like to invite the resubmission of a significantly-revised version that takes into account the reviewers' comments.

We cannot make any decision about publication until we have seen the revised manuscript and your response to the reviewers' comments. Your revised manuscript is also likely to be sent to reviewers for further evaluation.

Sincerely,

Yaakov Koby Levy

Academic Editor

PLOS Computational Biology

Nir Ben-Tal

Section Editor

PLOS Computational Biology

Reviewer's Responses to Questions

**Comments to the Authors:**

Reviewer #1: Disulfide-constrained peptides represent a promising peptide modality for drug design and therapeutic applications. However, their limited foldability poses challenges for library design, which is essential for discovering functional peptides. Cai and colleagues hypothesized that specific sequence patterns within these peptide scaffolds play a critical role in their spontaneous folding into stable topologies. Consequently, these sequences should be avoided during randomization in the initial library design. To create highly diverse disulfide-constrained peptide (DCP) libraries while preserving the inherent foldability of each scaffold, the researchers generated a large-scale dataset using yeast surface display combined with shotgun alanine scanning experiments. This dataset was used to train a machine-learning model based on a technique employed in natural language understanding. The authors validated the model through experimental methods, demonstrating that it can accurately predict the foldability of peptides across a broad range of sequences and identify residues critical for foldability. Building on these findings, the authors designed a new peptide library based on a de novo-designed disulfide-constrained peptide scaffold optimized for enhanced folding efficiency. Screening this library against various protein targets yielded peptides with good binding affinity. This study is likely to interest researchers in the fields of peptide design and drug discovery. I recommend the publication of this work, pending the resolution of the following issues.

1. It is unclear whether the BiFC assay can distinguish between natively-folded peptide and the disulfide-mispaired fold. Thus, at least two or three model sequences with different foldability should be selected to validate the BiFC assay.

2. HPLC chromatograms showing the oxidative folding of the peptide binders identified from library screening should be provided.

3. It would be interesting to experimentally characterize or computationally predict the structures of the identified peptide binders. A comparison between the structures of the identified peptides and that of the precursor scaffold should be conducted and discussed.

Reviewer #2: This work aims to improve peptide foldability for high-throughput yeast or phage display by constructing a sequence foldability database and training a BERT-based classification model. Why the model and data are of some value, peptide structures can now be predicted using AF2 or similar methods and peptide sequences can be designed using inverse programs. Large scale display approaches are of less value and should be combined with these kind of predictive models.

Some specific questions:

1. One of the key postulates in this work is that any library member capable of successful surface display on yeast cells must fold into a well-defined 3D structure, which may be true for well folded proteins, but for “peptides” that are normally flexible, it might be very different. An issue here is that what is the definition of “peptide” that the authors referred to. For common understandings, peptides usually do not have well folded structures. However, during the validation of generalizability, the protein with PDB code 1BH4, which is primarily composed of loops with various conformations, was selected. Does this conflict with the postulate? Additionally, the ES scores are near zero for both experimental results and predicted values. Does this imply that no residue is crucial for folding?

2. Sequence generation and diversity. How many sequences are generated from the shotgun alanine scanning? Additionally, what is the sequence diversity?

3. Detailed residue contribution. In Figures 1A, 2, and 4A, the ES scores of a continuous non-touchable peptide segment are close. How do you distinguish the detailed contribution of each residue?

4. Model adjustment. To achieve a higher F1-score, the model was adjusted from a ternary classification model to a binary classification model. Why did you adjust the ternary classification model instead of retraining the binary classification model from the original ratio data?

5. Visualization of Figure 2 data. The data in Figure 2 are too dense and the values are very similar. Please plot the model's prediction results as a bar chart and adjust the y-axis accordingly.

**Have the authors made all data and (if applicable) computational code underlying the findings in their manuscript fully available?**

Reviewer #1: Yes

Reviewer #2: None

PLOS authors have the option to publish the peer review history of their article (what does this mean?). If published, this will include your full peer review and any attached files.

Reviewer #1: No

Reviewer #2: No
---

## [Decision Letter · Decision Letter 1]

3 Nov 2024

Dear Dr. Zhang,

We are pleased to inform you that your manuscript 'Rapid prediction of key residues for foldability by machine learning model enables the design of highly functional libraries with hyperstable constrained peptide scaffolds' has been provisionally accepted for publication in PLOS Computational Biology.

Best regards,

Yaakov Koby Levy

Academic Editor

PLOS Computational Biology

Nir Ben-Tal

Section Editor

PLOS Computational Biology

Feilim Mac Gabhann

Editor-in-Chief

PLOS Computational Biology

Jason Papin

Editor-in-Chief

PLOS Computational Biology

Reviewer's Responses to Questions

**Comments to the Authors:**

Reviewer #1: The authors have revised the manuscript based on my comments. I recommend the publication of this manuscript as it is.

**Have the authors made all data and (if applicable) computational code underlying the findings in their manuscript fully available?**

Reviewer #1: None

PLOS authors have the option to publish the peer review history of their article (what does this mean?). If published, this will include your full peer review and any attached files.

Reviewer #1: No

---

## [Editor Report · Acceptance letter]

12 Nov 2024

PCOMPBIOL-D-24-01005R1 

Rapid prediction of key residues for foldability by machine learning model enables the design of highly functional libraries with hyperstable constrained peptide scaffolds

Dear Dr Zhang,

I am pleased to inform you that your manuscript has been formally accepted for publication in PLOS Computational Biology. Your manuscript is now with our production department and you will be notified of the publication date in due course.

With kind regards,

Marianna Bach
